# Meta-learning Optimizers for Communication-Efficient Learning

**Charles-Étienne Joseph**[*†♠]      *charles-etienne.joseph@mila.quebec*
**Benjamin Thérien**[*†♠]      *benjamin.therien@mila.quebec*
**Abhinav Moudgil**[‡♠]      *abhinav.moudgil@mila.quebec*
**Boris Knyazev**[◇]      *knyazevb@mila.quebec*
**Eugene Belilovsky** [‡,♠]      *eugene.belilovsky@concordia.ca*
*Department of Computer Science and Operation Research,*
*Université de Montréal, Montréal, Canada* †

*Department of Computer Science and Software Engineering,*
*Concordia University, Montréal, Canada* ‡

*Mila, Montréal, Canada* ♠

*Samsung - SAIT AI Lab, Montréal, Canada* ◇

**Reviewed on OpenReview:** *https://openreview.net/forum?id=uRbf9ANAns*

## Abstract

Communication-efficient variants of SGD, specifically local SGD, have received a great deal of interest in recent years. These approaches compute multiple gradient steps locally on each worker, before averaging model parameters, helping relieve the critical communication bottleneck in distributed deep learning training. Although many variants of these approaches have been proposed, they can sometimes lag behind state-of-the-art adaptive optimizers for deep learning. In this work, we investigate if the recent progress in the emerging area of learned optimizers can potentially close this gap in homogeneous data and homogeneous device settings while remaining communication-efficient. Specifically, we meta-learn how to perform global updates given an update from local SGD iterations. Our results demonstrate that learned optimizers can substantially outperform local SGD and its sophisticated variants while maintaining their communication efficiency. Our learned optimizers can even generalize to unseen and much larger datasets and architectures, including ImageNet and ViTs, and to unseen modalities such as language modeling. We therefore show the potential of learned optimizers for improving communication-efficient distributed learning.

## 1 Introduction

Rapidly training large-scale deep learning models is a problem of continued interest in the community. It requires a great deal of distributed computing resources that are often challenging to efficiently utilize. In many distributed learning settings, the communication overhead associated with distributed SGD can lead to inefficient use of computing resources and increased wall clock times (Lin et al., 2018). This reliance on frequent communication is especially impractical for training large models over heterogeneous hardware (Yuan et al., 2022). Moreover, it can increase the cost and complexity of designing data centers and other infrastructure to support the heavy communication constraints.

The primary communication overhead of distributed SGD comes from the synchronization of gradients computed by different workers. A recently popular direction to alleviate this overhead is local SGD (Stich, 2019), where each worker computes multiple ($H$) gradient steps independently before aggregating the weights

---

*Equal contribution

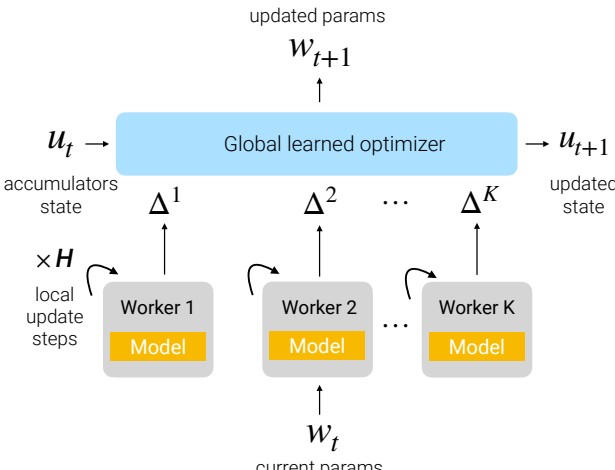

Figure 1: In local SGD, workers take $H$ local update steps (i.e., without communicating gradients) of SGD before communicating local parameter deltas ($\Delta^k$). This effectively reduces the number of communication steps by a factor $H$. Instead of averaging deltas at communication steps, we meta-train a global learned optimizer to aggregate the deltas into a more effective update.

(or deltas $\Delta^k$) of their local models (fig. 1). This reduces the communication costs. There is an algorithmic connection here to federated averaging McMahan et al. (2016), an analogous algorithm in the federated learning setting. The only difference between the two are assumptions about the workers computational capabilities, the data, and worker participation. We consider the setting of homogeneous high-resource workers, homogeneous data split among the workers, and full worker participation. We, therefore, refer to Local SGD (Stich, 2019) as the main prior work upon which we build as they also consider this setting.

Local SGD, however, has a number of challenges limiting its practical use. As the number of local steps $H$ increases the local models may diverge from each other leading to a degradation of performance (Wang et al., 2019). Local SGD also introduces a complex dynamic between the local and global updates, which can for example lead to complex interactions between hyperparameters such as global and local learning rates (Reddi et al., 2020).

Learned optimization through meta-learning has been an increasingly important topic of research interest (Andrychowicz et al., 2016). Advances have been made in scalable architectures (Wichrowska et al., 2017; Metz et al., 2022a), meta-learning strategies (Vicol et al., 2021) and the diversity and scale of meta-learning tasks (Metz et al., 2022b). Notably, Metz et al. (2022a) analyzed different learned optimizers in a large-scale study and introduced a highly efficient and simple per-parameter MLP optimizer and strong gradient-based features. However, these recent learned optimizers have not been studied in a communication-efficient distributed setting.

In this work, we propose learned optimization as an approach to alleviate the challenges of communication-efficient distributed learning. Specifically, we follow the setup of local SGD Stich (2019) with homogeneous devices and homogeneous data split among them and demonstrate that our global learned optimizers (fig. 1) meta-trained for this setting can outperform Local SGD and SlowMo (Wang et al., 2019) as well as data-parallel Adam and SGD. Our main contributions are:

- We demonstrate, for the first time, that learned optimizers can be used to improve local SGD for communication-efficient distributed learning, outperforming strong baselines and maintaining benefits even for a high number of local steps.

- We propose and evaluate two architectures for the learned optimization of local SGD, a worker-aware optimizer (LAgg-A) and a worker-invariant optimizer (LOpt-A), from which one can choose depending on the use-case.

- We demonstrate that our learned optimizers, even when meta-learned on a single or few architecture and dataset combinations, can generalize to new and much larger datasets and architectures, including ImageNet, ResNets, Vision Transformers (ViTs), and new modalities such as language modeling, obtaining competitive results in communication-efficient distributed settings.

## 2 Related Work

### 2.1 Local SGD and Communication-efficient DL

Local SGD has been analyzed in a number of works (Stich, 2019; Lin et al., 2018) which demonstrated that it both theoretically and empirically can lead to communication savings. It has also been shown that local SGD, particularly when combined with phases of regular SGD, can lead to better generalization (Lin et al., 2018) depending on the task scale (Ortiz et al., 2021).

Wang et al. (2019) introduced SlowMo using global or server-side momentum and showed that it can accelerate local SGD as well as a number of decentralized and asynchronous stochastic algorithms. A closely related algorithm has been proposed and extensively used in federated learning for communication efficiency (McMahan et al., 2017; Li et al., 2019). Work in this field has largely focused on addressing the heterogeneity of data across workers or clients (Karimireddy et al., 2020; Mishchenko et al., 2022). Most of the aforementioned algorithms struggle to handle large numbers of local steps H due to client drift, even in the non-federated setting. To address this, Douillard et al. (2024) proposes to use AdamW as the client side optimizer and Nesterov momentum to aggregate the deltas, allowing scaling to 500 local steps. These advancements are generally achieved by hand-designed algorithmic enhancements, whereas our approach relies on more flexible and potentially more powerful learnable mechanisms that may generalize these and more complex algorithms.

Another approach to communication-efficient learning is to compress the gradients or parameters. Two popular strategies in this setting are sparsification (Stich et al., 2018; Shi et al., 2019) and quantization (Alistarh et al., 2017) of the gradient. These strategies have also been combined by Wang et al. (2023). This line of work is thus orthogonal but complementary to our proposal. Communication efficiency has also been studied in the decentralized setting (Nabli & Oyallon, 2022; Nabli et al., 2023; Lian et al., 2018). Our work focuses on the centralized training setting but the methods can also be extended to decentralized training.

### 2.2 Learning to Optimize (L2O)

The idea of learning to learn and meta-learning has a long history (Schmidhuber, 1992; Thrun & Pratt, 2012). Many early works in this area focused on learning to efficiently acquire general knowledge or inductive bias. Hochreiter et al. (2001) proposed to use meta-learning in direct combination with gradient-based optimization to learn a separate network, which can be seen as a learned optimizer, which performs updates on another network. Andrychowicz et al. (2016) extended these ideas to a more scalable LSTM-based per-parameter architecture and demonstrated that the learned optimizer can generalize to new problems.

A large number of follow up works have improved L2O methods (Wichrowska et al., 2017; Metz et al., 2019; Chen et al., 2020; Metz et al., 2020; Harrison et al., 2022; Lv et al., 2017) (see Chen et al. (2022); Amos (2022) for surveys). These methods introduced different types of hierarchy into the learnable optimizer while simplifying its architecture in favor of stronger predefined features to improve its efficiency (Metz et al., 2022a). However, compared to our work, these have not considered a distributed setting, where learnable optimizers may significantly improve local SGD which is challenging to combine with adaptive optimizers.

Ji et al. (2019) proposed to learn the aggregation of gradients from workers in a distributed learning framework with a recurrent network. However, the focus was on improving non-local SGD while our work focuses on the communication efficiency in settings where each worker returns a message computed from multiple update steps. Furthermore, our approach is shown to generalize to new architectures and datasets.

## 3 Background

### 3.1 Local SGD

We consider a distributed training setup with $K$ workers. In local SGD (Stich, 2019), each communication round $t$ consists of $H$ local SGD steps performed independently on all $K$ workers. Each local step $h$ uses a minibatch of size $B_{\text{loc}}$. After completing the local updates, a global update is computed by averaging the local weight deltas and subtracting the average from previous weights, as shown on line 9 of algorithm 1.

---

**Algorithm 1:** Learned optimizers vs Local SGD. Steps used in both algorithms are not colored.

---

| **Input:** | $T$ | Number of communication steps |
| | $K$ | Number of workers |
| | $H$ | Number of local steps |
| | $\gamma$ | Local learning rate |
| | $\boldsymbol{W}_{0,0}$ | Initial weights |
| | $\mathcal{D}$ | Dataset |
| | $\mathcal{L}$ | Loss function |
| | $F_\phi$ | Learned optimizer |
| | $\boldsymbol{U}_0$ | Initial accumulators state |

1: **for** $t = 0$ to $T - 1$ **do**
2:   **for** $k = 0$ to $K - 1$ **in parallel do**
3:     **for** $h = 0$ to $H - 1$ **do**
4:       $X_h^{(k)}, Y_h^{(k)} \leftarrow \text{GET\_MINIBATCH}(\mathcal{D})$
5:       $\boldsymbol{W}_{t,h+1}^{(k)} \leftarrow \boldsymbol{W}_{t,h}^{(k)} - \gamma \nabla_{\boldsymbol{W}} \mathcal{L}\left(X_h^{(k)}, Y_h^{(k)}; \boldsymbol{W}_{t,h}^{(k)}\right)$
6:     **end for**
7:     $\Delta_t^{(k)} \leftarrow \boldsymbol{W}_{t,0}^{(k)} - \boldsymbol{W}_{t,H}^{(k)}$             // Difference in weights after $H$ local steps
8:   **end for**
9:   $\Delta_t \leftarrow \frac{1}{K} \sum_k \Delta_t^{(k)}$
10:   $\mathbf{A}_t, \boldsymbol{U}_{t+1} \leftarrow \text{ADA}(\boldsymbol{W}_{t,0}, \boldsymbol{U}_t, \Delta_t)$           // Compute Ada features and update state

         LAgg-A (§4.1.1):   $\boldsymbol{W}_{t+1,0} \leftarrow F_\phi\left(\mathbf{A}_t, \Delta_t^{(0,1,\ldots,K-1)}\right)$
11:         LOpt-A (§4.1.2):   $\boldsymbol{W}_{t+1,0} \leftarrow F_\phi\left(\mathbf{A}_t, \Delta_t\right)$       // Compute global update
    Local SGD Stich (2019):   $\boldsymbol{W}_{t+1,0} \leftarrow \boldsymbol{W}_{t,0} - \Delta_t$
12: **end for**

---

### 3.2 Learned Optimizer Input: Ada Features

In the learning-to-optimize and optimization literature, it is common to maintain accumulators of gradient (or in our case $\Delta_t$) statistics. For instance, Adam (Kingma & Ba, 2017) maintains per-parameter accumulators for the first and second moment of the gradient and Adafactor (Shazeer & Stern, 2018) maintains column-wise and row-wise sums of these moments. Although it is rarely done for hand-designed optimizers, it is possible to leverage more information by maintaining multiple ($> 2$) accumulators of these statistics at different time scales (e.g. multiple momentums with different coefficients). In fact, recent work (Metz et al., 2022a) develops a set of features based on this idea for training learned optimizers. Drawing inspiration from existing adaptive optimizers that often maintain accumulators similar to $\boldsymbol{a}_t$, we henceforth refer to these features as "Ada features" in the absence of a prior name.

We will now describe how we adapt Ada features to the local SGD setting. Ada features maintain three different per-parameter momentum accumulators ($m_{t,i}$) and one variance accumulator ($v_t$). In addition, they also maintain six accumulators of the column-wise ($c_{t,i}$) and row-wise ($r_{t,i}$) mean of the gradient. The

accumulator update is given as follows:

$$
\begin{aligned}
m_{t,i} &= \beta_i m_{t-1,i} + (1-\beta_i)\Delta_t & i \in \{1,2,3\}, \\
v_t &= \beta_4 v_{t-1} + (1-\beta_4)\Delta_t^2, \\
r_{t,i} &= \beta_i r_{t-1,i} + (1-\beta_i)\,\mathrm{ROW\_MEAN}(\Delta_t^2), & i \in \{5,6,7\}, \\
c_{t,i} &= \beta_i c_{t-1,i} + (1-\beta_i)\,\mathrm{COL\_MEAN}(\Delta_t^2), & i \in \{5,6,7\}, \\
\boldsymbol{U}_t &:= [m_{t,1}, m_{t,2}, m_{t,3}, v_t, r_{t,5}, r_{t,6}, r_{t,7}, c_{t,5}, c_{t,6}, c_{t,7}].
\end{aligned}
\tag{1}
$$

Here, we slightly abuse notation and define $\boldsymbol{U}_t$ to be the entire accumulator state for all parameters in the optimizee (column-wise and row-wise features are repeated for notational convenience). Note that Metz et al. (2022a) use $\nabla_t$ instead of $\Delta_t$ since they train learned optimizers in a fully-centralized setting. We propose using $\Delta_t$, the average local weight delta, instead of the gradient in our local SGD setting. After updating these accumulators at each communication step, one must compute additional learned optimizer input features to be concatenated with the accumulator values ($\boldsymbol{U}$) and an 11-dimensional timestep embedding ($\boldsymbol{T}$). The learned optimizer input-feature matrix $\boldsymbol{A}_t$ is created as follows:

$$
\begin{aligned}
\boldsymbol{T}_t &= [\tanh\left(\frac{t}{x}\right) \text{ for } x \in \{1,3,10,30,100,300,1000,3000,10000,30000,100000\}], \\
\boldsymbol{R}_t &= \left[\frac{1}{\sqrt{r_{t,5}}}, \frac{1}{\sqrt{r_{t,6}}}, \frac{1}{\sqrt{r_{t,7}}}, \frac{1}{\sqrt{c_{t,5}}}, \frac{1}{\sqrt{c_{t,6}}}, \frac{1}{\sqrt{c_{t,7}}}, \frac{m_{t,1}}{\sqrt{v}}, \frac{m_{t,2}}{\sqrt{v}}, \frac{m_{t,3}}{\sqrt{v}}, \frac{1}{\sqrt{v}}\right], \\
\boldsymbol{H}_t &= [m_{t,1}r_{t,5}c_{t,5},\ m_{t,2}r_{t,6}c_{t,6},\ m_{t,3}r_{t,7}c_{t,7},\ \Delta_t r_{t,5}c_{t,5},\ \Delta_t r_{t,6}c_{t,6},\ \Delta_t r_{t,7}c_{t,7}], \\
\boldsymbol{A}_t &= \boldsymbol{W}_t \odot \boldsymbol{U}_t \odot \boldsymbol{T}_t \odot \boldsymbol{H}_t \odot \boldsymbol{R}_t.
\end{aligned}
$$

Where $\odot$ denotes matrix concatenation across the feature dimension, $\boldsymbol{T}_t$ are time embeddings, $\boldsymbol{R}_t$ are reciprocal features, $\boldsymbol{H}_t$ are adafactor normalized features, and $\boldsymbol{W}_t$ are the current parameters of the optimizee. In algorithm 1, $\boldsymbol{A}_t$ is computed as follows:

$$
\mathbf{A}_t, \boldsymbol{U}_{t+1} = \mathrm{ADA}(\boldsymbol{W}_t, \boldsymbol{U}_t, \Delta_t).
\tag{2}
$$

Note that the complete matrix of input Ada features (Metz et al., 2022a) is given by $\Delta_t \odot \boldsymbol{A}_t$, however, we leave out $\Delta_t$ in the equations above to make explicit the differences of our proposed optimizers with respect to $\Delta_t$. We provide further details about these features in section A of the appendix.

## 4 Methodology

Our method builds upon local SGD. After $H$ local steps, we employ a per-parameter learned optimizer $F_\phi$ based on (Metz et al., 2022a) to compute the updated centralized weights (algorithm 1). By computing the centralized update using a small MLP, $F_\phi$, that is much more expressive than hand-designed updates, our method can be seen as a generalization of existing update methods such as taking the average iterate (Stich, 2019) or computing server-side momentum updates (Wang et al., 2019).

### 4.1 Learned Optimizer Training and Architectures

We consider the meta-learning framework with a learned optimizer $F_\phi$ parameterized by $\phi$ used to optimize a model with parameters $\boldsymbol{W}$. In the meta-learning formulation, $\phi$ is obtained by solving the following optimization problem:

$$
\min_\phi \ \mathbb{E}_{(\mathcal{D},\mathcal{L},\boldsymbol{W}_0)\sim\mathcal{T}} \left[ \mathbb{E}_{(X,Y)\sim\mathcal{D}} \left[ \frac{1}{TKH} \sum_{t=0}^{T-1} \sum_{k=0}^{K-1} \sum_{h=0}^{H-1} \mathcal{L}(X,Y; F_\phi(\cdot), \boldsymbol{W}) \right] \right].
\tag{3}
$$

For simplicity we remove the subscripts inside the sum term, but note that the exact value of the summands does depend on $t, k$, and $h$. Here, $\mathcal{T}$ is a distribution over optimization tasks defined as tuples of dataset $\mathcal{D}$,

objective function $\mathcal{L}$, and initial weights $\boldsymbol{W}_0$ associated with a particular neural architecture, $\phi$ represents the parameters of the learned optimizer, $H$ is the number of local steps, $K$ is the number of workers, and $T$ is the number of communication steps which we write as a fixed quantity for simplicity. In practice, during meta-training, $T$ is varied according to a truncation schedule (Metz et al., 2022a). We provide extended details of the meta-training process for our global learned optimizers in section B of the appendix.

In our experiments, $F_\phi$ is a two hidden layer 32 hidden dimension MLP with ReLU activations mapping the input Ada features for each parameter, $p$, in the optimizee to a two-dimensional vector, $[d_\phi, m_\phi]$. At step $t$, the learned optimizer update for all $p$ is given as follows:

$$F_\phi(\boldsymbol{A}_p \odot [\Delta_{t,p}]) = [d_{\phi,p}, m_{\phi,p}];$$
$$p_t = p_{t-1} - \lambda_1 d_{\phi,p} e^{(\lambda_2 m_{\phi,p})}. \tag{4}$$

Where $\boldsymbol{A}_p$ are the ada features computed from statistics of $p$ and $\lambda_1$ and $\lambda_2$ are constants set to 0.001. We drop the timestep subscript for some values of $p$ to avoid making the notation cumbersome. We propose two variants of our global learned optimizers, LAgg-A and LOpt-A (algorithm 1). LAgg-A takes advantage of individual deltas from all the workers and so can learn better optimizers when the number of workers is known and fixed beforehand. LOpt-A operates on the averaged delta, thus it is more versatile as it can be applied to the setting with an arbitrary number of workers, however, it can be less powerful than LAgg-A in certain cases as we show empirically.

### 4.1.1   Worker-aware Optimizer (LAgg-A)

Our first learned optimizer takes advantage of pre-aggregated information from each worker. Specifically, it takes as input $\Delta_t^{(1)}, \ldots, \Delta_t^{(k)}$ along with the Ada features computed from $\Delta_t$ (the average of $\Delta_t^{(1)}, \ldots, \Delta_t^{(k)}$). We refer to it as a *learned aggregator* (LAgg-A) as it learns to aggregate the workers' weight updates. With its access to pre-aggregated information, LAgg-A can learn complex interactions between workers potentially making more powerful weight updates. However, it requires fixing the number of workers $K$ before training, which in our experience is not an essential problem because oftentimes the distributed training assumes some standard fixed budget of workers.

### 4.1.2   Worker-invariant Optimizer (LOpt-A)

Our second proposed learned optimizer directly takes $\Delta_t$, the average of the updates from all workers, as an input feature along with the Ada features computed from it. This process is analogous to existing learned optimization proposed by Metz et al. (2022a) where the role of the gradient is replaced with $\Delta_t$. The advantage of LOpt-A versus LAgg-A is that it has the same number of parameters ($|\phi|$) regardless of the number of workers $K$. This can be useful when the same learned optimizer is applied for settings with variable $K$. However, this approach is less powerful as it cannot take advantage of individual deltas from all the workers.

### 4.2   Practical Considerations and LOpt-A and LAgg-A Overhead

As discussed by Reddi et al. (2020) the class of local algorithms can be described with a server-side optimizer and worker-side optimizer. For example, SlowMo (Wang et al., 2019) can be interpreted as adding momentum to the server optimization. Our design of algorithm 1 is such that the learned optimizer lives entirely on the server-side, making its use more practical and scalable than in non-communication-efficient settings.

Specifically, standard learned optimizers have an overhead of memory and compute. The memory must store state information and intermediate activations of the learned optimizer. In the case of our learned optimizer, this overhead (Metz et al., 2022a) is only incurred at the aggregation stage and can therefore live entirely on the server if one is available. Similarly, while the computational cost of the forward pass of learned optimizers provides a substantial overhead compared to simple add and multiply operations of SGD and Adam, in the case of our global learned optimizers this cost becomes small with respect to the large amount of data processed on workers during local updates. For instance, in table 1, we show that the overhead of our optimizers relative to Local SGD shrinks as the number of local steps grows. The figure reports total timings

for the forward and backward pass and optimizer step of Local SGD, LOpt-A, and LAgg-A for training ResNet50 (a) and ViT-Base (b) on 224 and 128 sized ImageNet, respectively. All timings are measured when training across K=8 A6000 GPUs. We further expand on this point in section 1 of the appendix.

| Local Steps | fwbw + Opt. Step Time (s) | | | Local Steps | fwbw + Opt. Step Time (s) | | |
|---|---|---|---|---|---|---|---|
| | Local SGD | LOpt-A | LAgg-A | | Local SGD | LOpt-A | LAgg-A |
| H=8 | 0.58 | 0.64 | 0.68 | H=8 | 0.57 | 0.65 | 0.71 |
| *overhead* | 1.00 | 1.11 | 1.17 | *overhead* | 1.00 | 1.14 | 1.25 |
| H=16 | 2.05 | 2.13 | 2.16 | H=16 | 2.11 | 2.19 | 2.25 |
| *overhead* | 1.00 | 1.04 | 1.05 | *overhead* | 1.00 | 1.04 | 1.07 |
| H=32 | 3.93 | 4.02 | 4.04 | H=32 | 4.19 | 4.25 | 4.32 |
| *overhead* | 1.00 | 1.02 | 1.03 | *overhead* | 1.00 | 1.01 | 1.03 |
| H=64 | 7.71 | 7.83 | 7.86 | H=64 | 8.51 | 8.60 | 8.66 |
| *overhead* | 1.00 | 1.02 | 1.02 | *overhead* | 1.00 | 1.01 | 1.02 |

(a) ViT-Base ImageNet-128       (b) ResNet50 ImageNet-224

Table 1: **The overhead of our optimizers relative to Local SGD shrinks to ZERO as the number of local steps grows.** We report total timings for the forward and backward pass and optimizer step of Local SGD, LOpt-A, and LAgg-A for training ResNet50 (a) and ViT-Base (b) on 224 and 128 sized ImageNet, respectively. All timings are measured when training across K=8 A6000 GPUs. All reported values are the median time in seconds of the final 40 training steps across 3 different training restarts. We provide more details in section C of the appendix.

## 5 Experiments

Learned optimizers are a relatively recent area and the experiments are usually run on small-scale datasets due to the challenges of meta-training and applying learned optimizers (Metz et al., 2022a). However, local SGD and its variants are typically studied in a large-scale distributed setup (Wang et al., 2019). Therefore, compared to the previous learned optimizers literature, we not only perform small-scale experiments but also experiment with larger and stronger architectures such as ViTs, including larger datasets such as ImageNet (Russakovsky et al., 2015) and more modalities such as language modeling (LM1B (Chelba et al., 2013)). All our experiments in the main manuscript assume a distributed setting with homogeneous devices and homogeneous data split among them. Following the convention in learned optimization Metz et al. (2022a; 2019); Harrison et al. (2022), we mainly report training loss for simplicity comparing among different optimizers. However, we do demonstrate that models trained by our global learned optimizers compare favorably on held-out data to models trained with other optimizers (see Figures 4 and 6).

### 5.1 Experimental Details

In the following two sections, we detail the training and evaluation tasks (optimizees) and the optimizers that we compare. We note that our experiments use standard datasets and evaluation protocols in learned optimization (Metz et al., 2022a). Our method is currently implemented in simulation. We meta-train and evaluate using 1 NVIDIA A100. For the presented results, each curve is an average over 10 trials with different seeds. Shaded regions represent one standard error from the mean.

#### 5.1.1 Datasets

We use the Fashion MNIST (FMNIST) dataset (Xiao et al., 2017) (10 classes) with $28 \times 28$ images. We also use the CIFAR-10 dataset (Krizhevsky et al., 2009) (10 classes) with $32 \times 32$ images. Finally, we scale our setup to the ImageNet dataset (Russakovsky et al., 2015) (1000 classes) with downsampled $32 \times 32$ and $64 \times 64$ images. We designate the dataset as ImageNet$^+$ when the larger images are used. For the language modeling task, we use LM1B (Chelba et al., 2013).

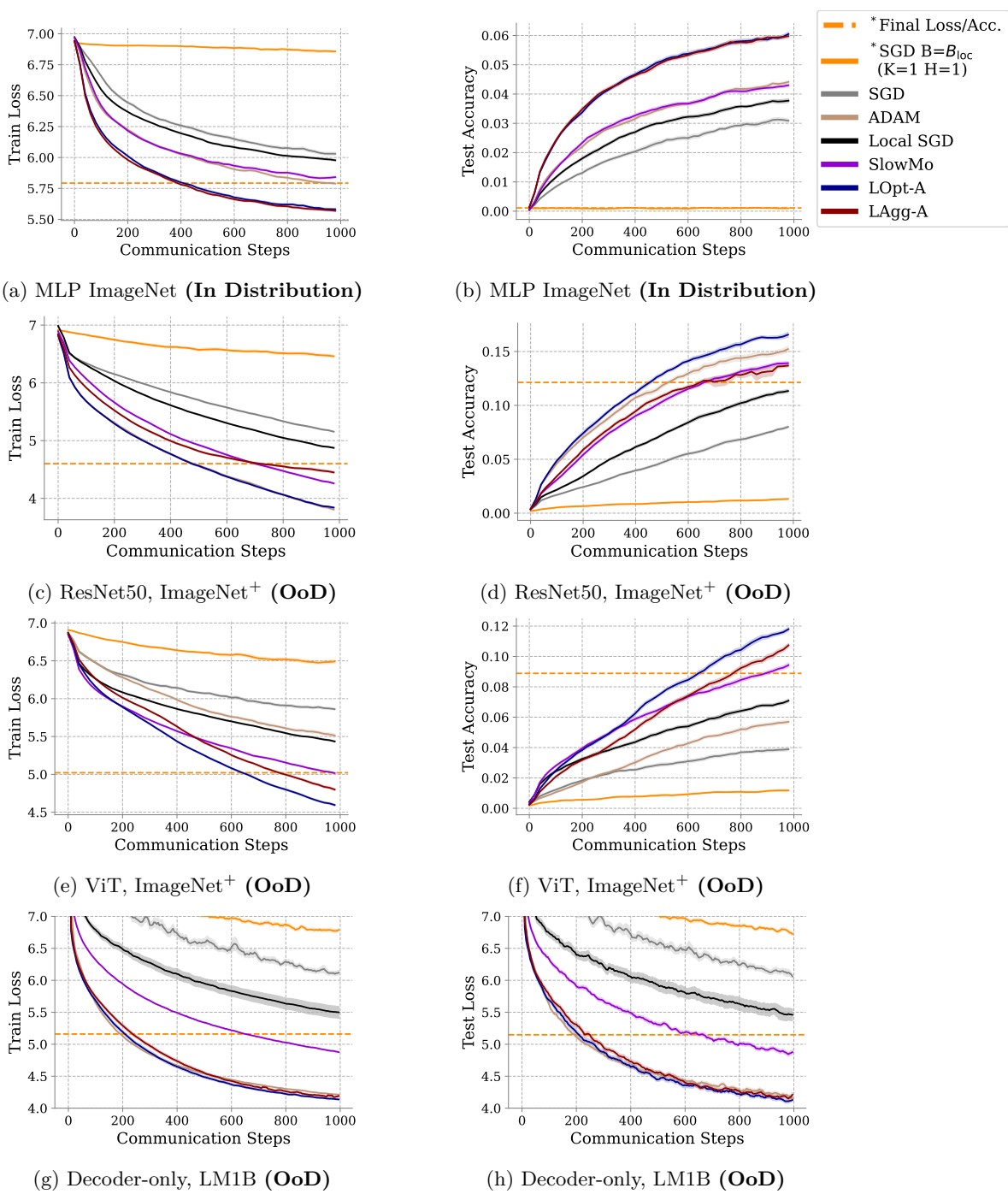

Figure 2: **Meta-Generalization to ResNet50, ViT, and a decoder-only language model.** We report train loss and test performance for each task. Note that the amount of data seen for ImageNet training, $K \cdot H \cdot 1000 \cdot B_{loc}$, corresponds to roughly 3.2 epochs, so the trained models do not reach convergence. In Figures (a,b), Our LAgg-A and LOpt-A optimizers, meta-trained on a 3-layer MLP (0.5M params) $32 \times 32$ ImageNet (IN32) task for 1000 steps, outperform extensively tuned baselines on the in-distribution task. Moreover, these optimizers generalize to IN64 on ResNet50 (c,d) ($50\times$ larger) and ViT (e,f) ($10\times$ larger). Finally, we also show (g,h) that the optimizers are useful for training a decoder-only transformer language model ($38\times$ larger). The orange SGD baseline is trained for $K \cdot H \cdot 1000$ optimization steps with a batch size $B_{loc} = 128$. The dotted orange line corresponds to the final performance of this baseline.

### 5.1.2 Neural Architectures

As for neural network architectures that our learned optimizers are going to optimize, we use multilayer perceptron (MLP) of two different sizes, both with ReLU activations. The first has two layers of 128 hidden nodes each and we refer to it as 2-Layer MLP. The second has three hidden layers of 128 hidden nodes each and we refer to it as 3-Layer MLP. We also use a convolutional neural network (CNN) of 3 layers with ReLU activations. All 3 layers have convolution kernels of size $3 \times 3$ and use "same" padding. The first layer has 32 units and stride 2, while the two other layers have 64 units and stride 1. We refer to this architecture as CNN. We also use standard architectures such as ResNet50 (He et al., 2016), a ViT equivalent in size to DeiT tiny (Touvron et al., 2021), and for the language task a decoder-only transformer with hidden size 192, 12 heads, and 12 layers.

### 5.1.3 Meta-training LOpt-A and LAgg-A

To meta-train our learned optimizers we estimate gradients using Persistent Evolution Strategies (PES) (Vicol et al., 2021) and take gradient descent steps using AdamW and a linear warmup plus cosine decay schedule. Each gradient is estimated from a batch of 8 tasks[1] each unrolled to a specific number of steps $T$. $T$ varies from 100 to 1000 during training according to a log-uniform truncation schedule. In our experiments, gradients are estimated with respect to the optimizee's training loss, except for the curves in fig. 4 whose gradients were estimated with respect to the optimizee's validation loss. During meta-training, the learning rate is warmed up for 100 steps to a maximum learning rate before being decayed (following a cosine decay schedule) to 1/3 of the maximum value. All the meta-training details are provided in appendix B.

Since our learned optimizers are meta-trained for 1000 steps, any evaluations beyond this training horizon is out-of-distribution. Therefore, we will evaluate our optimizers within their meta-training horizon of 1000 steps, which may not lead to convergence on all tasks, especially those requiring longer training (e.g., ImageNet).

### 5.1.4 Non-local SGD Baselines

We follow the setup of SlowMo (Wang et al., 2019) and provide a comparison to non-local algorithms. To do so, we train models using **SGD** (Robbins, 1951) and **Adam** (Kingma & Ba, 2017) for a number of steps equivalent to the total number of communication rounds used for the local methods. At each step, these baselines compute updates using the same effective batch size $K \times H \times B_{loc}$ as the local optimizers they are compared to. We also include an **SGD** baseline that uses $K \cdot H \times 1000$ steps and batch size $B_{loc} = 128$ in select experiments (figures 2, 3, and 10). The hyperparameters of these optimizers and details of their grid search are provided in appendix D.

### 5.1.5 Local SGD-based Baselines

Since our method focuses on improving server-side optimization and other client improving methods are orthogonal to our work, we provide two sufficient communication-efficient distributed baselines: local SGD (Stich, 2019) and SlowMo (Wang et al., 2019). An extensive hyper-parameter search is conducted for each baseline in every configuration. We detail the search process and report the best hyperparameters in appendix D. For each task, we use a local batch size $B_{loc}$ of 128.

### 5.2 Evaluating LAgg-A and LOpt-A In-distribution

In this section, we evaluate our proposed optimizers on FMNIST, CIFAR-10, and ImageNet using $H = 4$ iterations and $K = 8$ workers. Following the evaluation protocol of Metz et al. (2022a), in each case, we meta-train on a task (dataset and architecture pair) and perform evaluation on a new seed. That is, in distribution evaluations test the generalization of the optimizer to a new initialization of the model and new ordering of the data. Results reported in table 2 show that our learned optimizers, when evaluated in-distribution, enjoy faster convergence than local SGD and consistently outperform SlowMo. Figure 2

---

[1]Note that unless otherwise stated in our experiments all the tasks in a batch correspond to the same dataset and architecture, but different initial weights (see section section 4.1 for details).

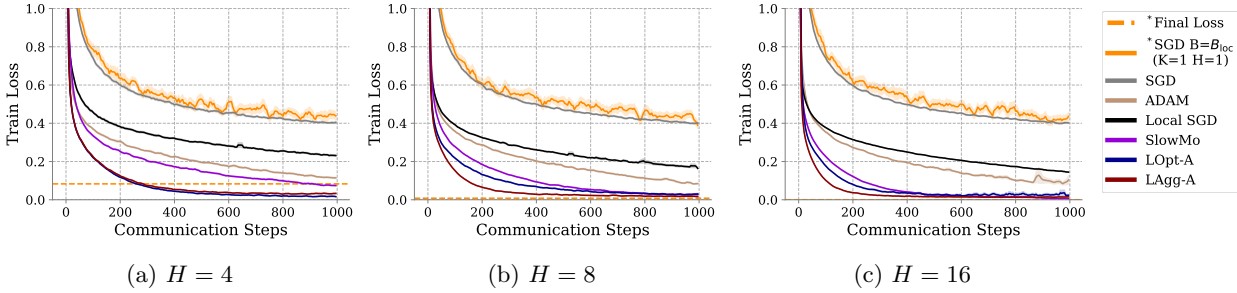

Figure 3: **LAgg-A outperforms all optimizers for $H \in \{4, 8, 16\}$ local steps**. All training curves are for FMNIST 2-Layer MLP. The orange SGD baseline is trained for $K \cdot H \cdot 1000$ optimization steps with a batch size $B_{loc} = 128$. The dotted orange line corresponds to the final performance of this baseline.

(a,b) presents the training and accuracy curves for ImageNet 3-Layer MLP. Figure 9, in appendix E, shows the training curves for FMNIST 2-Layer MLP (fig. 9a) and CIFAR-10 CNN (fig. 9b). We observe that LAgg-A and LOpt-A consistently converge faster than all other baselines from the start of training. Note that SlowMo is well-tuned and represents a very competitive approach in the class of methods that perform local updates (Wang et al., 2019).

Table 2: **Speedup with respect to local SGD**; reported as the ratio of the number of communications required by local SGD to the number of communications required by the other optimizer to achieve local SGD's minimum training loss (higher is better). In-distribution denotes that LOpt-A/LAgg-A are trained on the same task as the evaluation task. For meta-generalization, LOpt-A/LAgg-A are trained on ImageNet 3-layer MLP. A hyphen (–) indicates that the local SGD's minimum loss value was not achieved in the training run (1000 communication steps). When taking averages, hyphens are ignored.

| Optimizer | In-distribution | | | Meta-generalization | | | AVG |
|---|---|---|---|---|---|---|---|
| | FMNIST MLP | CIFAR10 MLP | ImageNet MLP | ImageNet ResNet50 | ImageNet ViT | LM1B Transformer | |
| Local SGD | 1.00 | 1.00 | 1.00 | 1.00 | 1.00 | 1.00 | 1.00 |
| Adam | $3.11 \pm 0.42$ | $1.69 \pm 0.16$ | $2.00 \pm 0.15$ | $2.10 \pm 0.06$ | $2.18 \pm 0.04$ | $3.56 \pm 0.09$ | 2.44 |
| SlowMo | $4.10 \pm 0.38$ | $4.81 \pm 0.74$ | $2.48 \pm 0.09$ | $1.99 \pm 0.08$ | – | $1.26 \pm 0.02$ | 2.93 |
| LOpt-A | $8.47 \pm 0.72$ | $\mathbf{9.62} \pm 0.36$ | $4.26 \pm 0.41$ | $\mathbf{2.56} \pm 0.12$ | $\mathbf{2.31} \pm 0.12$ | $\mathbf{6.76} \pm 0.43$ | **5.66** |
| LAgg-A | $\mathbf{9.80} \pm 0.54$ | $7.69 \pm 0.70$ | $\mathbf{4.63} \pm 0.26$ | $2.48 \pm 0.09$ | $1.88 \pm 0.08$ | $6.02 \pm 0.56$ | 5.42 |

## 5.3 Effect of Local Iterations ($H$)

We now analyze our learned optimizers' capability to scale to a larger number of local iterations ($H$). Specifically, we vary $H \in \{4, 8, 16\}$ and meta-train our learned optimizers on the FMNIST 2-Layer MLP task for each case (note that generalization to different $H$ is possible as we show in fig. 7). We report the performance of corresponding tuned baselines with the equivalent batch size (fig. 3). We also show the communication efficiency compared to local SGD and SlowMo in table 3. We observe that even for relatively high $H$ (Lin et al., 2018) there is an improvement over the strong communication-efficient baselines. As expected, table 3 illustrates higher $H$ yields more rapid convergence on a per communication step basis (due to more samples being processed). We also observe that LAgg-A begins to show a substantial advantage compared to LOpt-A at this higher $H$ value. We believe that using information from all the $\Delta_t^{(k)}$ allows for LAgg-A to learn a non-trivial aggregation scheme (compared to averaging), meaning it outperforms LOpt-A when the local models drift as $H$ gets higher.

In their recent work, Douillard et al. (2024) proposes DiLoCo, a new algorithm for the low-communication datacenter training of LLMs. They use AdamW as the client side optimizer and Nesterov momentum to

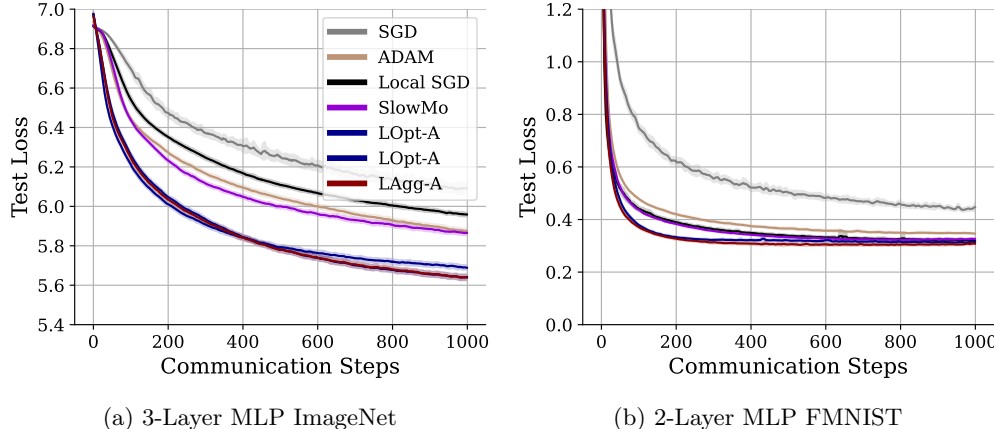

(a) 3-Layer MLP ImageNet

(b) 2-Layer MLP FMNIST

Figure 4: **Directly targeting validation loss during meta-training obtains strong performance on the test set.** Hand-designed optimizers were hyper-parameter-tuned to the validation set, while LAgg-A and LOpt-A were meta-trained to optimize validation loss on their respective tasks. We observe that learned optimizers trained to optimize validation loss during meta-training generalize seamlessly to the test set in our communication-efficient setting.

aggregate worker deltas, allowing scaling to $H = 2000$ local steps. However, they show that the step-to-loss improvement from using more local steps diminishes for $H > 100$. While our experiments show that LAgg-A and LOpt-A can improve the efficiency of local SGD for up to $H = 16$ steps, increasing H further will eventually become limited due to using SGD as the client-side optimizer. However, LAgg-A and LOpt-A are orthogonal to the choice of client-side optimizer and could easily be combined with AdamW on the client side to enable training with very large $H$ values at increased efficiency compared to DiLoCo.

Table 3: **Communication rounds until achieving 0.2 loss value for different optimizers at different H values** (lower is better).

| Optimizer | H=4 | H=8 | H=16 |
|---|---|---|---|
| Local SGD | – | 721 | 625 |
| SlowMo | 311 | 182 | 121 |
| LOpt-A | **119** | 121 | 89 |
| LAgg-A | 122 | **81** | **55** |

## 5.4 Outer Loop Generalization

Following conventions in the learned optimization literature (Metz et al., 2022b;a) our focus in this work has been demonstrating the efficient convergence of the learned optimizer. Thus in our experiments, the outer loop of the meta-learning problem (see equation in section 4.1) evaluates the training data. In this section, we demonstrate that we can also obtain strong performance on the validation data using our learned optimizer. Figure 4 reports the test loss of learned optimizers meta-trained using the validation loss objective and baselines tuned using validation loss on 3-Layer MLP ImageNet (fig. 4a) and 2-Layer MLP FMNIST (fig. 4b). We observe similar trends to our training loss plots (figs. 2, 3 and 10). In fig. 4a, we observe that both LAgg-A and LOpt-A converge significantly faster and obtain lower final test loss than the baselines. In fig. 4b, LAgg-A and LOpt-A converge faster than other baselines reaching a test loss around iteration 200 that baselines only reach after 600 iterations of training. While both plots show similar relative trends between our learned optimizers and the baselines, we note that they represent distinct scenarios: in fig. 4a the model is far from convergence, while in fig. 4a the model converges and is close to overfitting. Our learned optimizers handle both situations gracefully.

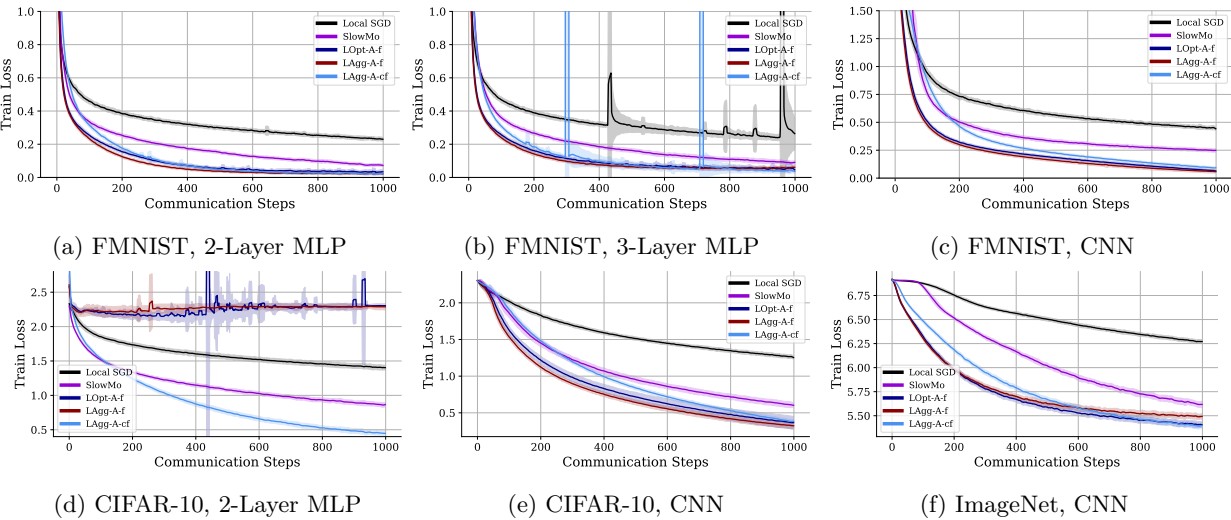

Figure 5: **Meta-generalization to new datasets and new architectures**. All optimizers were meta-trained and hyper-parameter tuned for task 5a. Meta-generalization is evaluated in three progressively more difficult settings: new architectures same dataset (5b, 5c), new dataset same architecture (5d), and new dataset and new architecture (5e, 5f). Learned optimizers achieve strong generalization to different architectures on the same dataset, but experience difficulties optimizing the same architecture on a new dataset. However, the improvements of performance from LAgg-A-f to LAgg-A-cf in plot (5f) shows that these issues can be mitigated by scaling training tasks. Finally, both learned optimizers evaluated generalize outside of the training data distribution and architecture in plots (5e) and (5f).

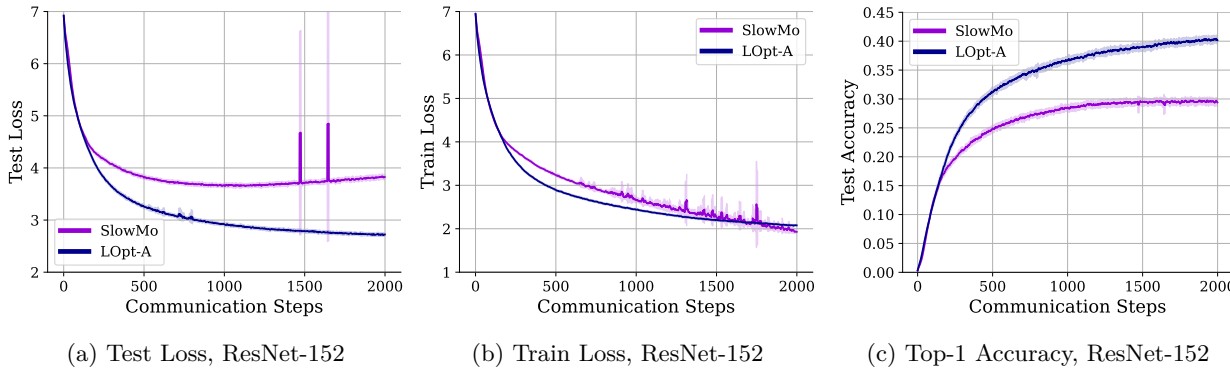

Figure 6: **Training ResNet-152 on Imagenet64**. We meta-train a new LOpt-A optimizer for 2000 communication steps and compare it to a well-tuned Slowmo baseline.

## 5.5 Meta-generalization

The results are reported in figs. 2, 5 and 7. In fig. 5, we evaluate generalization in three progressively more difficult settings: new architectures same dataset (figs. 5b and 5c), new dataset same architecture (fig. 5d), and new dataset and new architecture (figs. 5e and 5f). In fig. 7, we evaluate the capability of our learned optimizers trained at one $H$ value to generalize to another. Both these figures report results from optimizers meta-trained and hyperparameter tuned on FMNIST and are therefore of smaller scale. In contrast, fig. 2 reports larger scale experiments showing the generalization performance of optimizers meta-trained and hyperparameter tuned on ImageNet to much larger tasks, such as ResNet50, ViT, and a decoder-only transformer which are especially relevant in deep learning.

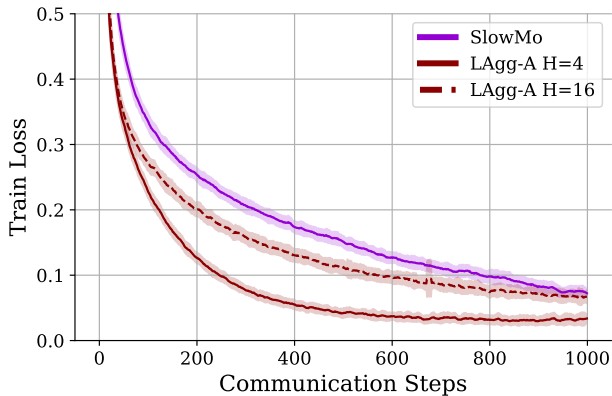

Figure 7: **LAgg-A trained at** $H = 16$ **generalizes to** $H = 4$**.** We observe that **LAgg-A H=16** trained at $H = 16, K = 8$ improves upon a strong SlowMo baseline at $H = 4, K = 8$.

### 5.5.1 Meta-trained on FMNIST and CIFAR-10

In fig. 5, **LAgg-A-f** and **LOpt-A-f** are trained on the FMNIST, 2-Layer MLP task, while **LAgg-A-cf** is trained on a two-dataset task using FMNIST and CIFAR-10 with the same 2-Layer MLP. All baseline models use hyperparameters tuned on the FMNIST 2-Layer MLP task. Every model is trained using $K = 8$ and $H = 4$ with the exception of **LAgg-A H=16** (trained using $K = 8$ and $H = 16$).

### 5.5.2 Generalization to Unseen Architectures

We observe that our learned optimizers can generalize to unseen architectures (figs. 5b and 5c). In particular, LAgg-A-f trained on 2-Layer MLP tasks can perform well on a CNN and an MLP of different depth, showing generalization in our communication-efficient setting. Performance in the case of the CNN is particularly strong without having seen this task during training.

### 5.5.3 Generalization to Unseen Datasets

We observe that LAgg-A meta-trained on FMNIST 2-Layer MLP struggles to optimize the same architecture on CIFAR-10 (fig. 5d) and ImageNet (fig. 12 in appendix E.4). We note, however, that including an additional task (CIFAR-10, MLP) during meta-learning can significantly improve performance. Specifically, we observe that this learned optimizer (LAgg-A-cf) is able to generalize to both of its in-distribution tasks (CIFAR-10 and FMNIST MLP) as well as improve performance on ImageNet MLP. This suggests that stronger meta-generalization can be achieved by scaling the training tasks in our communication-efficient setting as has been demonstrated for standard optimization settings in the learned optimization literature (Metz et al., 2022b).

### 5.5.4 Generalization to Unseen Datasets and Architectures

Interestingly, we observe (figs. 5e and 5f) that both learned optimizers, LAgg-A-f and LAgg-A-cf achieve strong generalization when varying both the dataset (CIFAR-10 and ImageNet) and the architecture (CNN). This is perplexing when contrasted with the poor generalization observed in the previous paragraph when training the same MLP architecture on CIFAR-10 and ImageNet. We hypothesize that these difficulties arise from changes in MLP dimensions required to accommodate CIFAR-10 and ImageNet 3-channel images as compared to FMNIST's single-channel images. As for the strong performance when optimizing the CNN, we believe this is due to the architecture's inductive biases for image processing, making it relatively easier to optimize.

### 5.5.5 Scaling up: Meta-trained on ImageNet

We now consider a larger-scale meta-training task along with an array of target modern architectures and tasks. Figure 2 reports meta-generalization results to ResNet50, a ViT model, and a Decoder-only LM. Lagg-A and LOpt-A were meta-trained on the 3-layer MLP ImageNet task, while the baselines were extensively hyperparameter-tuned for this task.

Figure 2 (c,d) shows meta-generalization results for ResNet50 trained on ImageNet$^+$ ($64 \times 64$ images). We observe strong generalization of LOpt-A, outperforming all baselines, while LAgg-A performs well at the beginning, but encounters some instability later on in training. The generalization of LOpt-A is particularly notable as ResNet50 has $50\times$ more parameters than the architecture seen during meta-training and the input images are two times larger.

Figure 2 (e,f) shows meta-generalization results for a ViT model of the same size as DeiT tiny (Touvron et al., 2021). Both Lagg-A and LOpt-A show strong generalization, but LOpt-A performs best again. Interestingly, for this task, both global learned optimizers outperform the communication-efficient baselines by a large margin.

Figure 2 (g,h) shows meta-generalization to a decoder-only transformer for causal language modeling on LM1B. We observe notably strong generalization performance from both LAgg-A and LOpt-A, improving on all the baselines by a large margin. As shown in table 2 both optimizers reach the minimal loss value achieved by local SGD in over 6 times fewer communication steps.

These results establish the existence of highly promising meta-generalization capabilities Metz et al. (2022a;b) for learned optimizers in communication-efficient settings that appear to improve with scaling the meta-training task. Moreover, they demonstrate that such optimizers can also generalize to different values of $H$, suggesting that it is possible to obtain learned optimizers that are general in $H$ and in tasks by scaling training compute and task variety while using higher $H$ values.

### 5.5.6 Training ResNet-152 on ImageNet

To demonstrate that our method can smoothly train large models on ImageNet, we meta-train a learned optimizer on four 3-layer MLP ImageNet-32 classification tasks of width $w \in \{64, 128, 256, 512\}$ for 2000 steps in a $K = 8$ and $H = 8$ setting. We compare exclusively to a tuned SlowMo baseline since this has been the strongest performing baseline throughout the paper. For both optimizers, we use gradient clipping to a norm of 5 and weight decay of $1e-4$ for the local steps. Our results in Figure 6 demonstrate that LAgg-A outperforms SlowMo and reaches 40% top-1 accuracy after 2000 communication steps ($\approx 14$ epochs). While we do not test on more steps as this exceeds our meta-training widow, we expect that learned optimizers meta-trained for longer should achieve analogous strong performance.

## 6 Limitations

Despite our method's strong performance in a variety of settings, it still has some limitations. In some cases meta-generalization is limited (fig. 5d). This problem is generally observed in previous L2O literature and some recent methods (e.g. STAR (Harrison et al., 2022), Thérien et al. (2024)) can complement our method to further improve meta-generalization.

This work is a first step towards the use of L2O for communication-efficient learning and our focus in this work is on learning the global step of local learning schemes. We believe that that combining these methods with other communication-efficient techniques such as gradient sparsification (appendix E.5) may prove effective and have results going in this direction, but we leave a more detailed study of this to future work.

Finally, our method does not have local learnable components and relies on vanilla SGD steps performed locally. This was a design choice to allow for a more simple and scalable investigation. We leave room in future research to address these limitations.

# 7 Conclusion

We have demonstrated the utility of learned optimization for improving communication-efficient distributed training in homogeneous data and device settings. Specifically, we proposed two learned optimizer architectures for this setting — LAgg-A and LOpt-A. Our results illustrate that these optimizers can effectively be applied in communication-efficient distributed settings. We highlight their generalization capabilities to unseen architectures and datasets. These findings establish learned optimization as a promising direction for improving communication-efficient distributed training algorithms for deep learning while scaling to diverse architectures, datasets, and $H$ values. They also hold promise not only in the current context but also in decentralized and federated learning scenarios.

**Broader Impact Statement**

This paper proposes learned algorithmic enhancements for communication efficient optimizers. Although there are many potential future societal consequences of our work, the application of methods proposed herein does not directly pose societal consequences. That being said, in the long term, data-driven optimization algorithms do offer the possibility of creating inscrutable optimization algorithms which could be tailored to a user's specification and may not always be aligned with human values.

# Acknowledgements

We acknowledge support from Mila-Samsung Research Grant, FRQNT New Scholar [*E.B.*], the Canada Excellence Research Chairs Program in Autonomous AI, and the FRQNT Doctoral (B2X) scholarship [*B.T., A.M.*]. We also acknowledge resources provided by Compute Canada and Calcul Québec.

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

## A  Learned Optimizers Architecture and Features

Both our proposed global learned optimizers, LOpt-A and LAgg-A, are 2-hidden-layer MLPs with 32 hidden nodes per layer and ReLU activation functions. In addition to the input Ada features introduced in section 3.2, LAgg-A, has $K$ other input features: $\Delta_t^{(k)}$ for each of the $K$ workers. Therefore, LAgg-A has a total of $38 + K$ input features. LOpt-A, has single additional input feature: $\Delta_t$, the average of $\Delta_t^{(1)}, \ldots, \Delta_t^{(k)}$. All but the timestep features are normalized to have a second moment of 1 across the tensor. With all of this in mind we can compute the number of meta-parameters $\phi$ in the MLP for each of our learned optimizers. LOpt-A has a total of $|\phi| = 2402$ meta-parameters, while LAgg-A for values $K \in \{8, 16, 32\}$ respectively have $|\phi| \in \{2626, 2882, 3394\}$ meta-parameters.

Table 4: **Ada features used with our global learned optimizers.** All the coefficients, $\beta_i$, are learnable parameters adjusted during meta-optimization.

| Type | # | Description | Equation |
|---|---|---|---|
| **Accumulators** | 3 | Momentum accumulators with coefficients $\beta_i, i \in \{1, 2, 3\}$. | $m_{t,i} = \beta_i m_{t-1,i} + (1 - \beta_i)\Delta_t$ |
| | 1 | Second moment accumulator with coefficients $\beta_4$. | $v_t = \beta_4 v_{t-1} + (1 - \beta_4)\Delta_t^2$ |
| | 3 | Adafactor row accumulator with coefficients $\beta_i, i \in \{5, 6, 7\}$. | $r_{t,i} = \beta_i r_{t-1,i} + (1 - \beta_i)\,\texttt{row\_mean}(\Delta_t^2)$ |
| | 3 | Adafactor accumulator with coefficients $\beta_i, i \in \{5, 6, 7\}$. | $c_{t,i} = \beta_i c_{t-1,i} + (1 - \beta_i)\,\texttt{col\_mean}(\Delta_t^2)$ |
| **Accumulator Features** | 3 | Momentum values normalized by the square root of the second moment for $i \in \{5, 6, 7\}$. | $\frac{m_{t,i}}{\sqrt{v}}$ |
| | 1 | The reciprocal square root of the second moment value. | $\frac{1}{\sqrt{v}}$ |
| | 6 | The reciprocal square root of the Adafactor accumulators. | $\frac{1}{\sqrt{r_{t,i} \text{ OR } c_{t,i}}}$ |
| | 3 | Adafactor delta features for $i \in \{5, 6, 7\}$. | $\Delta_t \times r_{t,i} \times c_{t,i}$ |
| | 3 | Adafactor momentum features for $i, j \in \{(5, 1), (6, 2), (7, 3)\}$. | $m_{t,j} \times r_{t,i} \times c_{t,i}$ |
| **Time Features** | 11 | Time Features for $x \in \{1, 3, 10, 30, 100, 300, 1000, 3000, 10^4, 3 \cdot 10^4, 10^5\}$. | $\tanh\left(\frac{t}{x}\right)$ |
| **Parameters** | 1 | Parameter value. | $w_t$ |
| | 1 | Average Delta across workers. (LOpt-A only) | $\Delta_t = \frac{1}{K}\sum_{k=1}^K \Delta_t^{(k)}$ |
| | K | Per-worker Deltas. (LAgg-A only) | $\Delta_t^{(1)}, \ldots, \Delta_t^{(k)}$ |

## B  Meta-training Process

Algorithm 2 describes the meta-training of our global learned optimizers. We adapt the algorithms from Vicol et al. (2021) (algorithm 1) with minimal changes to notation for reader convenience. Note that we use PES to estimate gradients for improved meta-training efficiency. However, other algorithms than PES can be used to estimate the global learned optimizer's gradients. On lines 4 and 18 we sample the inner problem length log uniformly between $I_{\min}$ and $I_{\max}$, ensuring that the start of the inner-problem is seen more frequently during meta-training. The states $s^{(i)}$ contain information (e.g., the accumulator state $\boldsymbol{u}$, the iteration count, the onpimizee's weights $\boldsymbol{W}$, etc.) that must be propagated from one truncated unroll to the next. $\xi^{(i)}$ track perturbations across unrolls, $c$ tracks the total inner problem length, and $\hat{g}_{\text{PES}}$ is the PES gradient estimate. For simplicity, our meta-training algorithms only shows meta-training for a single task $(\boldsymbol{W}_0, \mathcal{D}, \mathcal{L})$ where

---

**Algorithm 2:** Meta-training our Global Learned Optimizers with PES. Note that this algorithm has been adapted from algorithm 1 of (Vicol et al., 2021) with minimal changes to the notation.

---

**Input:** Optimizer parameters $\theta_t$, gradients

1: **Input:**

  $s_0$     initial state
  $T_r$     truncation length for partial unrolls
  $K$     local workers
  $H$     local steps
  $N$     number of particles
  $\sigma$     standard deviation of perturbations
  $\alpha$     local learning rate
  $\mathcal{O}$     outer steps
  $I_{\min}$     minimum inner steps
  $I_{\max}$     maximum inner steps
  $\mathcal{L}$     training objective
  $L$     meta loss
  $\mathcal{D}$     Dataset
  $\theta$     Learned Optimizer Parameters
  unroll($\cdot$)   corresponds to Algorithm 1 with $T = T_r$

2:   Initialize $\theta \leftarrow \text{NeuralNetworkInit}()$
3:   Initialize $s^{(i)} \leftarrow s_0$ for $i \in \{1, \dots, N\}$
4:   Initialize $\xi^{(i)} \leftarrow 0$ for $i \in \{1, \dots, N\}$
5:   Initialize $\kappa^{(i)} \leftarrow \text{LOG\_UNIFORM}(I_{\min}, I_{\max})$ for $i \in \{1, \dots, N\}$
6:   Initialize $\boldsymbol{c} \leftarrow \boldsymbol{0}$
7:   **for** $j = 1, \dots, \mathcal{O}$ **do**
8:    $\hat{g}_{\text{PES}} \leftarrow 0$
9:    **for** $i = 1, \dots, N$ **do**
10:     $\epsilon^{(i)} \leftarrow \begin{cases} \text{draw from } \mathcal{N}(0, \sigma^2 I), & \text{if } i \text{ is odd} \\ -\epsilon^{(i-1)}, & \text{if } i \text{ is even} \end{cases}$
11:     $s^{(i)}, L_{T_r}^{(i)} \leftarrow \text{unroll}(s^{(i)}, \theta + \epsilon^{(i)}, \alpha, \mathcal{L}, T_r, K, H, \mathcal{D})$
12:     $\xi^{(i)} \leftarrow \xi^{(i)} + \epsilon^{(i)}$
13:     $\hat{g}_{\text{PES}} \leftarrow \hat{g}_{\text{PES}} + \xi^{(i)} L_{T_r}^{(i)}$
14:     $\boldsymbol{c} \leftarrow \boldsymbol{c} + T_r \cdot \boldsymbol{1}$
15:    **end for**
16:    $\hat{g}_{\text{PES}} \leftarrow \frac{1}{N\sigma^2} \hat{g}_{\text{PES}}$
17:    $\theta \leftarrow \theta - \text{ADAMW}(\hat{g}_{\text{PES}})$
18:    **for** $i = 1, \dots, N$ **do**
19:     **if** $\boldsymbol{c}_i \geq \kappa^{(i)}$ **then**
20:      $s^{(i)} \leftarrow s_0$
      $\xi^{(i)} \leftarrow 0$
      $\kappa^{(i)} \leftarrow \text{LOG\_UNIFORM}(I_{\min}, I_{\max})$
      $\boldsymbol{c}_i \leftarrow 0$
21:     **end if**
22:    **end for**
23:   **end for**

---

only the initialization of $\boldsymbol{W}_0$ is varied across resets of $s^{(i)}$ but not $\mathcal{D}$ or $\mathcal{L}$. The multitask setting is analogous and simply requires maintaining additional $(s^{(i)}, s^{(i)}, \kappa, c)$ for each task.

As stated in section 4.1, our meta-learning objective, $L_{T_r}$, is the average loss over $T_r$ iterations. This optimization problem usually requires long unrolls of the compute graph. In our study, we meta-train our global optimizers using $I_{\min} = 100$ and $I_{\max} = 1000$. The only exceptions to this are the optimizers trained

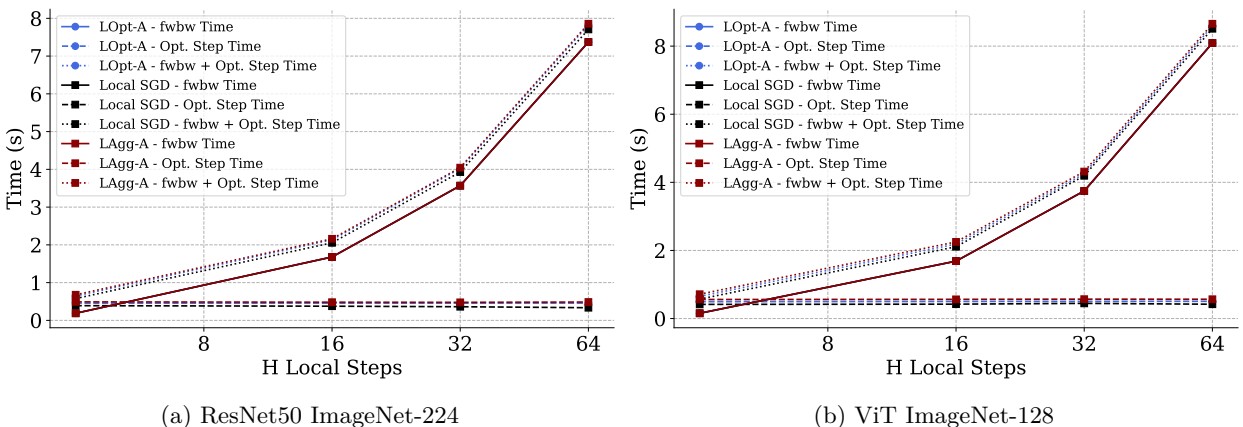

(a) ResNet50 ImageNet-224

(b) ViT ImageNet-128

Figure 8: **The overhead of our optimizers relative to Local SGD shrinks as the number of local steps grows.** We report timings for the forward and backward pass, optimizer step time, and total step time of local SGD, LOpt-A, and LAgg-A for training ResNet50 (a) and ViT-Base (b) on 224 and 128 sized ImageNet across K=8 GPUs. All reported values are the median timings of the final 40 training steps across 3 different training restarts.

in fig. 4 for which we found using a truncation schedule leads to overfitting later in optimizee training. Therefore, we opted to train these optmizers with $I_{min} = 1000$ and $I_{max} = 1000$ and the optimizer in section 5.5.6 which used $I_{min} = 100$ and $I_{max} = 2000$. For most of the learned optimizers in our study, we meta-trained for $\mathcal{O} = 5\ 000$ outer steps. The only exceptions are the learned optimizers used in section 5.3 that were meta-trained for $\mathcal{O} = 10\ 000$ outer steps. During meta-training, we used AdamW as our optimizer with a warmup cosine decay schedule. The learning rate starts at $3e-10$ and warms up linearly to the peak value of $3e-3$ after the first 100 iterations. It then decays to the final value of $1e-3$ until the end of meta-training.

## C Practical Considerations

Without any optimizations beyond `jax.jit` (e.g. no quantization of the LO or custom kernel), we compare the step time of LOpt-A and LAgg-A to Local SGD. Specifically, we report timings for the forward and backward pass, optimizer step time, and total step time of the optimizers when training ResNet50 and ViT-Base on 224 and 128 sized ImageNet across K=8 GPUs. In Table 1 of the main text, we report the per-step overhead of our learned optimizers relative to Local SGD. We observe that the overhead vanishes as the number of local steps grows because the cost of local steps dominates. Therefore, learned optimizers are quite appealing for low-communication optimization. In addition, figure 8 reports the explicit timings as the number of local steps is increased.

## D Baselines

For every configuration in which we used the baseline optimizers, namely the architecture, the dataset and the different values of $K$ and $H$, we ran an exhaustive hyperparameter sweep over the following values. For SGD and Adam, we searched over the learning rate $\alpha \in \{1, 5e-1, 1e-1, 5e-2, 1e-2, 5e-3, 1e-3, 5e-4, 1e-4, 5e-5, 1e-5\}$. For local SGD, we searched over the local learning rate $\gamma \in \{1, .5, .3, .1\}$. For SlowMo, we varied the local learning rate $\gamma \in \{1, 0.5, 0.3, 0.1\}$, the slow learning rate $\alpha \in \{1/\gamma, 5e-1/\gamma, 1e-1/\gamma, 5e-2/\gamma, 1e-2/\gamma, 5e-3/\gamma, 1e-3/\gamma, 5e-4/\gamma, 1e-4/\gamma, 5e-5/\gamma, 1e-5/\gamma\}$ and the momentum $\beta \in \{0.99, 0.95, 0.9, 0.85, 0.8, 0.75, 0.7, 0.65, 0.6, 0.55, 0.5\}$. The best hyperparameters for each configuration are regrouped in table 5.

Table 5: **Best hyperparameters for baselines**. When it is not specified, the tuning targets training loss.

| Configuration | Objective | SGD ($\alpha$) | Adam ($\alpha$) | Local SGD ($\gamma$) | SlowMo ($\gamma$ / $\alpha$ / $\beta$) |
|---|---|---|---|---|---|
| FMNIST, 2-Layer MLP, $K = 8$, $H = 4$ | Validation loss | 0.1 | 0.001 | 0.5 | 0.3 / 0.01 / 0.8 |
| FMNIST, 2-Layer MLP, $K = 8$, $H = 4$ | Train loss | 0.1 | 0.01 | 0.3 | 0.1 / 1 / 0.95 |
| FMNIST, 2-Layer MLP, $K = 8$, $H = 8$ | Train loss | 0.1 | 0.005 | 0.3 | 0.1 / 1 / 0.95 |
| FMNIST, 2-Layer MLP, $K = 8$, $H = 16$ | Train loss | 0.1 | 0.005 | 0.1 | 0.1 / 1 / 0.95 |
| FMNIST, 2-Layer MLP, $K = 16$, $H = 4$ | Train loss | 0.1 | 0.005 | 0.5 | 0.1 / 1 / 0.95 |
| FMNIST, 2-Layer MLP, $K = 32$, $H = 4$ | Train loss | 0.1 | 0.005 | 0.5 | 0.3 / 1.66 / 0.9 |
| CIFAR-10, CNN, $K = 8$, $H = 4$ | Train loss | 1 | 0.01 | 1 | 0.5 / 2 / 0.9 |
| ImageNet, 3-Layer MLP, $K = 8$, $H = 4$ | Train loss | 1 | 0.001 | 0.3 | 0.1 / 1 / 0.85 |

Table 6: **Best hyperparameters for $k = 1, H = 1, B = B_{loc}$ SGD baseline**. When it is not specified, the tuning targets training loss.

| Configuration | Objective | Steps (K*H*1000) | SGD ($\alpha$) |
|---|---|---|---|
| FMNIST, 2-Layer MLP, $K = 1$, $H = 1$ | Train loss | 32000 | 0.05 |
| FMNIST, 2-Layer MLP, $K = 1$, $H = 1$ | Train loss | 64000 | 0.05 |
| FMNIST, 2-Layer MLP, $K = 1$, $H = 1$ | Train loss | 128000 | 0.05 |
| ImageNet, 3-Layer MLP, $K = 1$, $H = 1$ | Train loss | 32000 | 0.05 |

# E Extended results

## E.1 Evaluating LAgg-A and LOpt-A in-distribution

As mentioned in section 5.2, we present the in-distribution training curves for FMNIST 2-Layer MLP and CIFAR-10 CNN in fig. 9.

## E.2 Effect of the Number of Workers ($K$)

In fig. 10 we evaluate the performance of our method as the number of workers ($K$) increases. Similarly to section 5.3, we vary $K \in \{8, 16, 32\}$. For each different value of $K$, we meta-train our learned optimizers on the FMNIST 2-Layer MLP task. We observe that our learned optimizers can gracefully handle more workers, reaching a lower loss in fewer iterations than all baselines by a significant margin in each case. While LAgg-A performs better, it needs to be retrained for each $K$. In contrast, LOpt-A does not have to be retrained. Therefore, each optimizer needs to be carefully chosen depending on the use-case.

## E.3 Ablating Ada Features

Our learned optimizers leverage powerful per-parameter optimization features proposed in Metz et al. (2022a). Here we investigate how important these are to the performance of the optimizers. Specifically, we consider directly feeding the $\Delta_t$ only or each $\Delta_t^{(k)}$ to the learned optimization MLP network without adding any of the Ada features described in appendix A. We denote these baselines as LOpt and LAgg, respectively (excluding the -A). We observe that a large improvement in convergence and training stability is obtained by using Ada features in both cases (fig. 11). However, we note that the performance of LOpt and LAgg alone still experiences improved convergence early in training with respect to local SGD. These baselines have no momentum calculations and the optimizer is an MLP (as opposed to a recurrent model) thus there is no way to maintain history information (unlike SlowMo's momentum). It is therefore notable that LAgg can achieve similar, albeit slower, convergence to SlowMo during the first 600 iterations. However, LAgg does seem to cause training instability from iteration 800 onwards. Interestingly, the models trained with Ada features do not suffer from such instabilities, despite being trained with the same schedule as LAgg, further demonstrating their benefit.

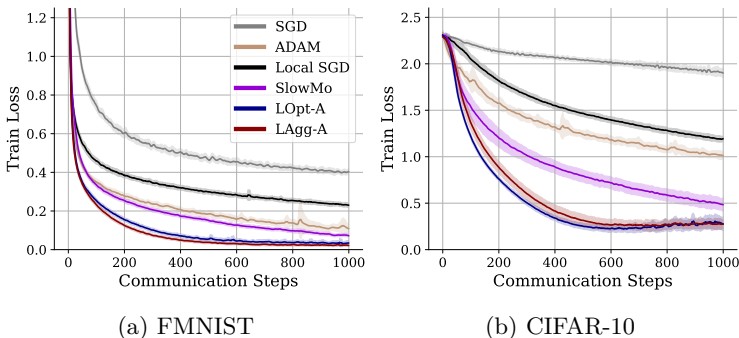

(a) FMNIST

(b) CIFAR-10

Figure 9: **Learned optimizers enable communication-efficient learning.** Our LOpt-A and LAgg-A outperform strong communication-efficient baselines such as SlowMo and local SGD. They also outperform well-tuned standard optimization strategies at equivalent effective batch sizes.

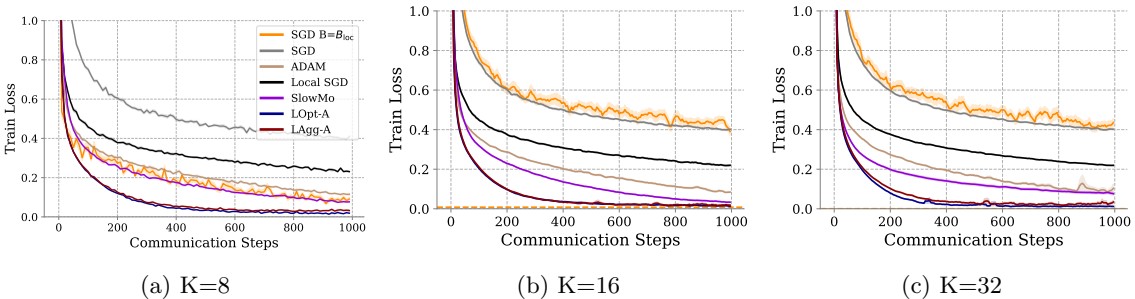

(a) K=8

(b) K=16

(c) K=32

Figure 10: **LAgg-A outperforms all optimizers for $K \in \{8, 16, 32\}$ workers**. All training curves are reported for the $28 \times 28$ FMNST dataset. The top row plots training curves for a small CNN, while the bottom row plots training curves for an MLP. All experiments use $H = 4$. The orange SGD baseline is trained for $K \cdot H \cdot 1000$ optimization steps with a batch size $B_{loc} = 128$. The dotted orange line corresponds to the final performance of this baseline.

### E.4 Meta-generalization

As mentioned in section 5.5.1, we present the meta-generalization results for the ImageNet 2-Layer MLP task here in fig. 12.

### E.5 Learned Optimization with Compressed Updates

As mentioned in section section 2.1, gradient or parameters compression techniques, such as gradient sparsification (Stich et al., 2018; Shi et al., 2019) are orthogonal approaches to reducing communication cost in distributed deep learning. We show that our method works in conjunction with gradient sparsification and compare it with baselines that are also applying sparsification. Specifically, we meta-train learned optimizers while implementing top-k sparsification for the deltas, using top-k values $\{1, 0.1, 0.01\}$ (fraction of the deltas that are communicated each step). Hyperparameters for the baselines can be found in table 7.

Figure 13 presents the performance achieved by our learned optimizers versus the allotted communication budget (in $\log_2$ bits). We observe that our learned optimizer achieves better training loss while communicating less. For example, LAgg-0.01 achieves a lower training loss than SlowMo-1, while having a much lower communication budget. Figure 14 shows the train loss achieved by of our learned optimizer during training for different top-k values. We can see that for each value, our learned optimizers achieve a lower training loss than the baselines. Finally, fig. 15 presents the effect of different top-k value on both our learned optimizers, LOpt-A and LAgg-A. Both our optimizers are robust to top-k values down to 0.01.

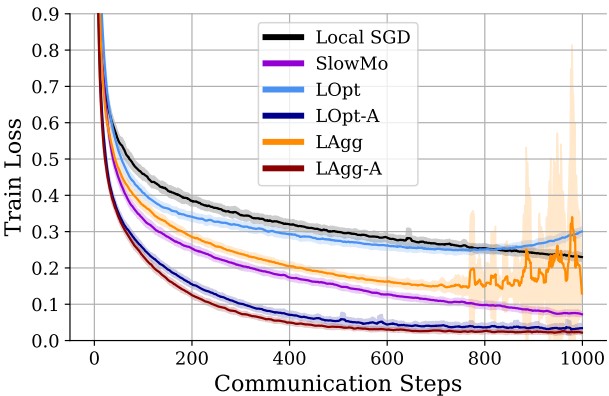

Figure 11: **Effect of Ada features on optimizer performance.** Each learned optimizer is trained and tested on FMNIST 2-Layer MLP at $H = 4$ and $K = 8$.

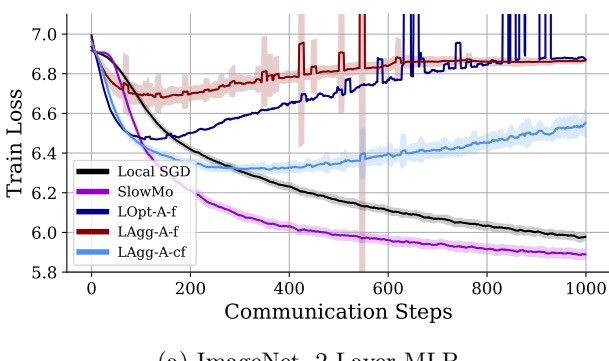

(a) ImageNet, 2-Layer MLP

Figure 12: **Meta-generalization to new datasets and new architectures**. All optimizers were meta-trained and hyper-parameter tuned for task 5a. The plot above shows generalization to ImageNet using the same 2-layer MLP architecture as when meta-training. We observe that our learned optimizers exclusively training on 2-layer MLP FMNIST fail to generalize to ImageNet, but that LAgg-A improves substantially when it was also meta-trained for CIFAR-10. This suggests that meta-generalization can be improved in our communication-efficient setting by simply adding more tasks.

All in all, we observe that our proposed learned optimization framework for distributed learning can similarly provide improvements for sparsification demonstrating that as in the non-learned setting, combining different local learning compression techniques, like sparsification and quantization (Basu et al., 2019), can further improve communication efficiency.

### E.6 Learning Optimizers for Federated Learning

Motivated by privacy and collaboration, federated learning (McMahan et al., 2016) allows a model to be collaboratively trained by any number of devices without centrally storing or even exchanging client data. Relative to centralized training, this setting has many difficulties. First, distributed and federated stochastic gradient descent algorithms, such as FedSGD (McMahan et al., 2016), have a high communication cost resulting from the synchronization gradients across all workers. To alleviate this, federated learning algorithms such as FedAvg (McMahan et al., 2016) allow clients to take multiple local steps with their local models before averaging the weights and updating the global model. This results in reduced communication cost and faster convergence per communication round. However, FedAvg can have convergence issues in part due to the local (per client) models diverging from each other (Karimireddy et al., 2020), a phenomenon known as client drift. Variants of FedAvg, such as FedAdagrad, FedYogi, and FedAdam (Reddi et al., 2020), or

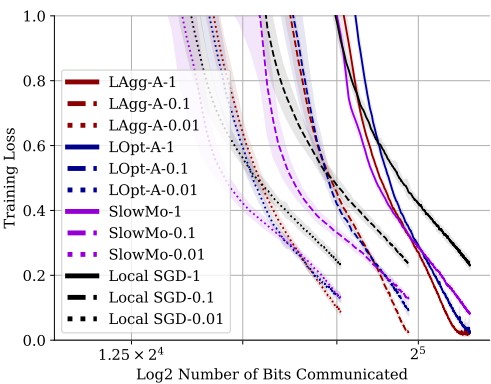

Figure 13: **Performance of top-k learned optimizers versus the communication budget.** We show train loss versus the $\log_2$ number of bits communicated. Full lines represent top-1, dashed lines show top-0.1 and dotted lines are top-0.01.

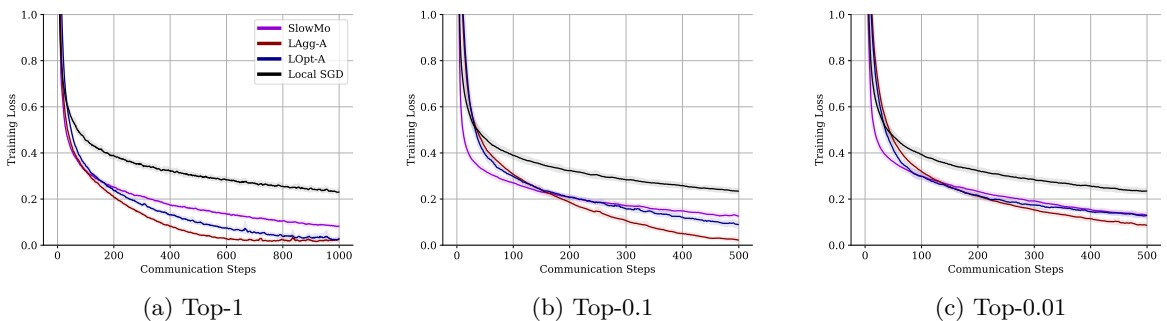

(a) Top-1          (b) Top-0.1          (c) Top-0.01

Figure 14: **Performance of top-k learned optimizers with different top-K values.** Our learned optimizer enjoy better performance than baselines for each value of top-k.

other techniques like SCAFFOLD (Karimireddy et al., 2020), which is directly targeting client drift, can improve the convergence speed and final accuracy. Another well-known problem within the FL setting is label-based data heterogeneity, also known as statistical heterogeneity, which can also affect client drift. Given the similarities of federated learning with local SGD, LOpt-A and LAgg-A can also be meta-trained in a FL setting.

### E.6.1 Methodology

We employ a per-parameter learned optimizer defined by the function $F_\phi$ to compute the global update. We designed a client invariant optimizer operating on $\Delta_t$ corresponding to LOpt-A, that we name FedLOpt. We use this learned optimizer to replace the averaging step in the FedAvg, as described in algorithm 3. The notable difference with algorithm 1 is that we sample a subset $\mathcal{K}$ of clients, each with their own data (either homogeneously or heterogeneously distributed depending on the experiment), which is expected for a federated learning setting.

### E.6.2 Dirichlet Partitioning

To simulate label-based heterogeneity among the client, we use Dirichlet partitioning (Li et al., 2021). This process is controlled by a hyperparameter $\alpha$ that controls the amount of data heterogeneity. A high value of $\alpha$ will induce homogeneity while a low value will create heterogeneity, as demonstrated in fig. 16.

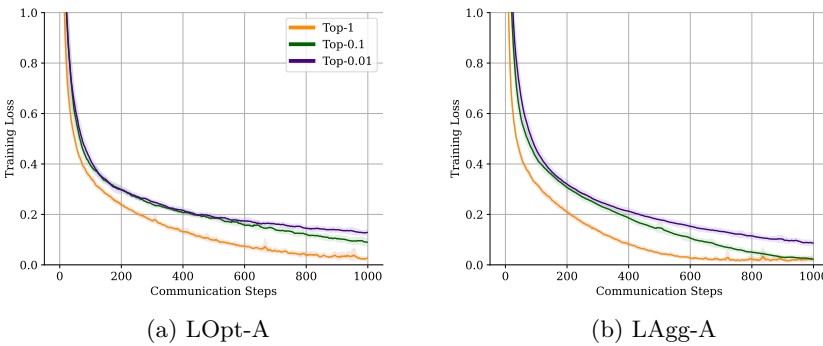

(a) LOpt-A  (b) LAgg-A

Figure 15: **Effect of the top-k value on the performance of learned optimizers.**

Table 7: **Baselines best hyperparameters for different top-k values.**

| Optimizer | Top-K Value | $\gamma$ | $\alpha$ | $\beta$ |
|---|---|---|---|---|
| SlowMo | 1 | 0.1 | 1 | 0.95 |
| SlowMo | 0.1 | 0.3 | 1.66 | 0.8 |
| SlowMo | 0.01 | 0.1 | 5 | 0.3 |
| Local SGD | 1 | 0.3 | - | - |
| Local SGD | 0.1 | 0.3 | - | - |
| Local SGD | 0.01 | 0.5 | - | - |

### E.6.3    Experiments

For our experiments, clients are sampled uniformly at random without replacement each round. New clients are selected each round. We use a total of 400 clients throughout the experiments with a participation rate of 0.025 for a total of $K = 10$ clients each round. Each client gets the same number of samples. We compute the number of local steps $H$ as described in algorithm 3 with $H = \frac{EB}{B_{loc}}$ where $E$ is the number of local epochs we want, $B$ is the total number of samples on each client and $B_{loc}$ is the local batch size used to compute one local step. For all experiments, we fix $E = 1$, unless specified otherwise, and $B_{loc} = 20$. For all experiments, we used $T = 1000$ rounds. We used 3 different initializations, each with a different random seed, and we report the average value and the shaded region in plots corresponds to one standard deviation. We use the data from all the test split of the dataset to compute the accuracy. We chose this metric as it is more common in federated learning literature.

We compare our learned optimizer to the following FL baselines: FedAvg (McMahan et al., 2016), FedAda-grad, FedYogi and FedAdam (Reddi et al., 2020), each combining FedAvg with a different form of adaptive optimization techniques. We fine-tune our baselines for each task with a grid search over the local learning rate $\eta_l$ and global learning rate $\eta_g$, using the validation loss as metric. We fix $\tau = 0.001$, $\beta_1 = 0.9$ and $\beta_2 = 0.99$ for all adaptive federated optimizers. We also compare our method with SCAFFOLD (Reddi et al., 2020), that differs from other FL optimizers by directly aiming at reducing the effect of client drift by introducing control variates in local updates. As was done in Reddi et al. (2020), we fix the global learning rate to 1 and sweep over different values of local learning rate. The values of the different hyperparameters can be found in table 8.

### E.6.4    Results

We use two tasks, namely FMNIST with MLP and CIFAR-10 with CNN. For both task, we evaluate two settings: one where the data is homogeneously distributed ($\alpha = 100$) and one where data is heterogeneous ($\alpha = 0.1$) among clients. Results are reported in figs. 17 and 18.

---

**Algorithm 3:** FedLOpt and FedAvg. Shared steps are not colored.

---

**Data:** Number of training rounds $T$; Number of workers $K$; Number of local steps $H$; Local learning rate $\gamma$; Initial weights $\boldsymbol{W}_{0,0}$; Loss function $\mathcal{L}$; Learned optimizer $F_\phi$; Initial accumulators state $\boldsymbol{u}_0$

**1** **for** $t \in \{0, 1, \ldots, T-1\}$ **do**

**2** $\quad$ Sample subset $\mathcal{K}$ of $K$ clients

**3** $\quad$ **for** $k \in \mathcal{K}$ **in parallel** **do**

**4** $\quad\quad$ **for** $h \in \{0, 1, \ldots, H-1\}$ **do**

**5** $\quad\quad\quad$ $X_h^{(k)}, Y_h^{(k)} \leftarrow \text{GET\_MINIBATCH}(\mathcal{D})$

**6** $\quad\quad\quad$ $\boldsymbol{W}_{t,h+1}^{(k)} \leftarrow \boldsymbol{W}_{t,h}^{(k)} - \gamma \nabla_{\boldsymbol{W}} \mathcal{L}\left(X_h^{(k)}, Y_h^{(k)}; \boldsymbol{W}_{t,h}^{(k)}\right)$

**7** $\quad\quad$ Difference in weights after $H$ local steps: $\Delta_t^{(k)} \leftarrow \boldsymbol{W}_{t,0}^{(k)} - \boldsymbol{W}_{t,H}^{(k)}$

**8** $\quad$ Averaging: $\Delta_t \leftarrow \frac{1}{K} \sum \Delta_t^{(k)}$

**9** $\quad$ Compute Ada features (§3.2) and update state: $\mathbf{A}_t, \boldsymbol{u}_{t+1} \leftarrow \text{ADA}(\boldsymbol{W}_{t,0}, \boldsymbol{u}_t, \Delta_t)$

**10** $\quad$ Compute global update:

$\quad\quad$ FedLOpt (§4.1.2): $\boldsymbol{W}_{t+1,0} \leftarrow F_\phi\left(\mathbf{A}_t, \Delta_t\right)$

$\quad\quad$ FedAvg (McMahan et al., 2016): $\boldsymbol{W}_{t+1,0} \leftarrow \boldsymbol{W}_{t,0} - \Delta_t$

---

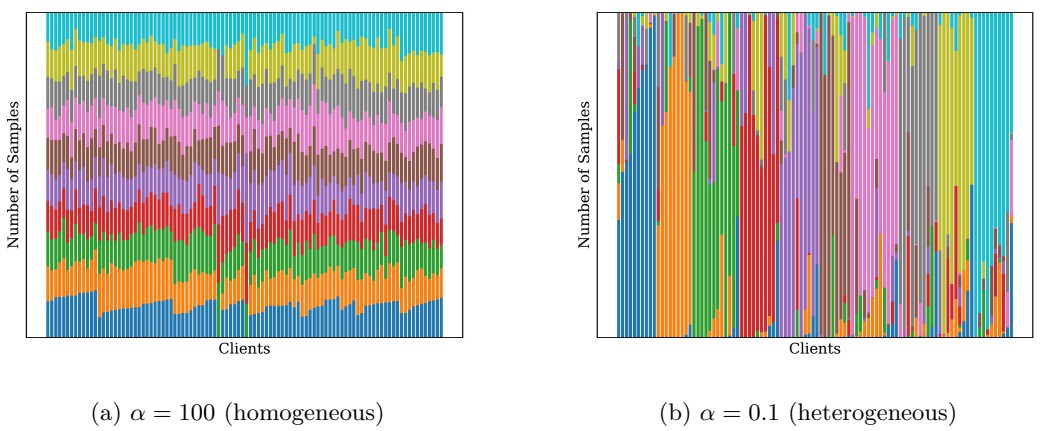

(a) $\alpha = 100$ (homogeneous) $\quad\quad\quad\quad\quad\quad\quad$ (b) $\alpha = 0.1$ (heterogeneous)

Figure 16: **Dirichlet partitioning of clients.** Each class label is represented by a different color. Figure 16a shows homogeneous data among clients. Figure 16b shows label-based data heterogeneity among clients.

Our learned optimizer achieves better test accuracy in the homogeneous and heterogeneous settings than all well-tuned baselines in our study. This is the case for both FMNIST MLP and CIFAR-10.

Table 8: **Best hyperparameters for federated baselines**

| Optimizer | Task | $\alpha$ | $\gamma$ | $\eta$ |
|---|---|---|---|---|
| FedAdam | FMNIST MLP | 100 | 0.1 | 0.001 |
| FedAdam | FMNIST MLP | 0.1 | 0.1 | 0.001 |
| FedAdam | CIFAR-10 CNN | 100 | 0.1 | 0.01 |
| FedAdam | CIFAR-10 CNN | 0.1 | 0.1 | 0.01 |
| FedYogi | FMNIST MLP | 100 | 0.1 | 0.001 |
| FedYogi | FMNIST MLP | 0.1 | 0.1 | 0.001 |
| FedYogi | CIFAR-10 CNN | 100 | 0.1 | 0.01 |
| FedYogi | CIFAR-10 CNN | 0.1 | 0.1 | 0.01 |
| FedAdagrad | FMNIST MLP | 100 | 0.1 | 0.01 |
| FedAdagrad | FMNIST MLP | 0.1 | 0.1 | 0.01 |
| FedAdagrad | CIFAR-10 CNN | 100 | 0.1 | 0.1 |
| FedAdagrad | CIFAR-10 CNN | 0.1 | 0.1 | 0.1 |
| FedAvg | FMNIST MLP | 100 | 0.1 | - |
| FedAvg | FMNIST MLP | 0.1 | 0.1 | - |
| FedAvg | CIFAR-10 CNN | 100 | 1 | - |
| FedAvg | CIFAR-10 CNN | 0.1 | 0.1 | - |
| SCAFFOLD | FMNIST MLP | 100 | 0.01 | 1 |
| SCAFFOLD | FMNIST MLP | 0.1 | 0.01 | 1 |
| SCAFFOLD | CIFAR-10 CNN | 100 | 0.05 | 1 |
| SCAFFOLD | CIFAR-10 CNN | 0.1 | 0.05 | 1 |

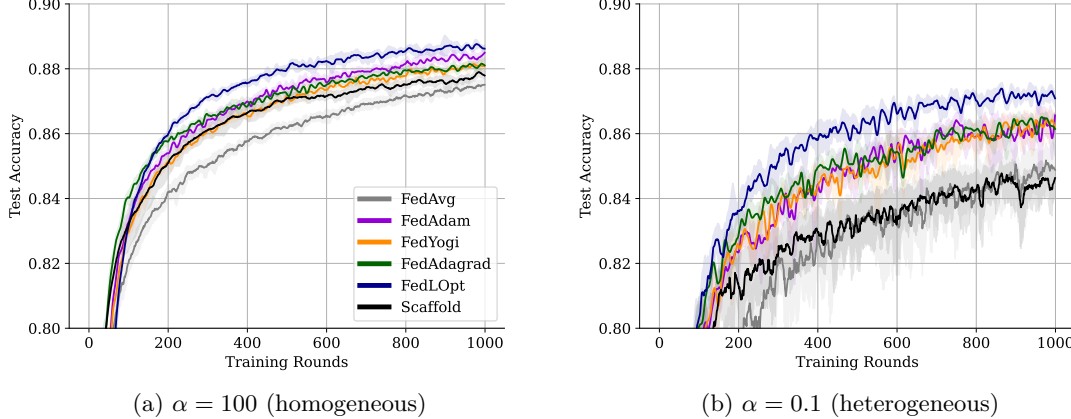

(a) $\alpha = 100$ (homogeneous)    (b) $\alpha = 0.1$ (heterogeneous)

Figure 17: **Evaluation on FMNIST MLP.** Both learned optimizers were meta-trained on the same task they are evaluated on.

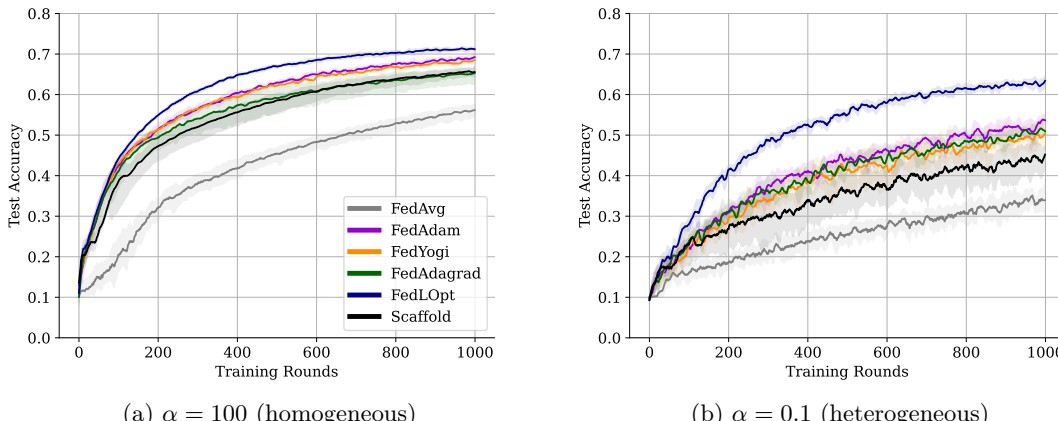

(a) $\alpha = 100$ (homogeneous)       (b) $\alpha = 0.1$ (heterogeneous)

Figure 18: **Evaluation on CIFAR-10 CNN.** Both learned optimizers were meta-trained on the same task they are evaluated on.

