# OpenReview forum: "Meta-learning Optimizers for Communication-Efficient Learning"
_TMLR — Accepted by TMLR_

### Review · Reviewer_y11Q · 2024-10-22

**Summary Of Contributions:**

This paper introduces a novel approach to communication-efficient distributed learning by applying meta-learning to optimize the aggregation of local updates in a distributed setting. The main contributions are:
1. Two learned optimizer architectures, LAgg-A and LOpt-A, designed to improve upon local SGD and its variants.
2. Demonstration of the learned optimizers' effectiveness across various tasks, including generalization to unseen architectures and datasets.
3. Empirical analysis of the optimizers' performance with different numbers of local steps (H) and workers (K).
4. Integration of the learned optimizers with gradient sparsification techniques.
5. Empirical evidence showing improved convergence rates and communication efficiency compared to standard optimizers and communication-efficient baselines.

The authors evaluate their method on various datasets (FMNIST, CIFAR-10, ImageNet) and architectures (MLPs, CNNs, ResNet50, ViT, and a decoder-only transformer), demonstrating both in-distribution performance and generalization capabilities.

**Audience:**

No

**Broader Impact Concerns:**

1. Generalization and Robustness: The paper shows some generalization capabilities, but it's crucial to ensure that these learned optimizers are robust across a wide range of scenarios to prevent unexpected behavior in critical applications.
2. Privacy Considerations: In distributed learning scenarios, especially federated learning, privacy is a key concern. The paper does not address how the proposed methods might interact with privacy-preserving techniques like differential privacy.

**Claims And Evidence:**

No

**Requested Changes:**

1. Mathematical rigor and clarity:
   - Clearly define all mathematical notation, especially A_t and F_φ
   - Provide precise formulation of the meta-learning objective, explaining dependencies on t and k
   - Fix reference formatting and ensure consistent mathematical terminology

2. Experimental evaluation:
   - Include experiments with larger models (>100M parameters)
   - Report standard accuracy metrics (Top1/Top5) for image classification tasks
   - Add comparisons with recent federated learning methods [14-20]
   - Use logarithmic scale for loss plots where appropriate
   - Conduct ablation studies to analyze the impact of different components

3. Practical considerations:
   - Analyze computational overhead, especially for large models
   - Test robustness to non-IID data distributions
   - Investigate behavior with partial worker participation

4. Implementation details:
   - Release source code

**Strengths And Weaknesses:**

Strengths:
1. Novel application of meta-learning to communication-efficient distributed optimization.
2. Comprehensive empirical evaluation across multiple datasets and architectures.
3. Demonstration of generalization capabilities to unseen tasks and larger models.
4. Integration with existing communication-efficient techniques (e.g., gradient sparsification).
5. Improvement over strong baselines like local SGD and SlowMo in terms of convergence speed and communication efficiency.

Weaknesses:

1. Limited theoretical analysis:
  The paper predominantly focuses on empirical results without providing theoretical foundations for the proposed learned optimizers, even in simple settings such as smooth, strongly convex functions [1,2,3,4,5]. While Section 4.1 presents the meta-learning framework and optimization problem, it lacks crucial theoretical guarantees on convergence or optimality.

2. Insufficient analysis of heterogeneous data scenarios:
  The paper assumes uniformly distributed data across workers, overlooking a critical aspect of real-world distributed learning applications. In federated learning settings, data heterogeneity is a fundamental challenge [6,7], with clients typically having non-IID (Independent and Identically Distributed) data distributions. The paper doesn't investigate how LAgg-A or LOpt-A perform under such heterogeneous conditions, such as when workers have significantly different data distributions or varying amounts of data. This omission is particularly significant given the extensive literature demonstrating the impact of data heterogeneity on distributed optimization [8,9,10] and the various methods developed to address these challenges.

3. LAgg-A's limited applicability in dynamic worker settings:
  As described in Section 4.1.1, LAgg-A's architecture depends on pre-aggregated information from each worker, making its input size proportional to the number of workers K. While the authors dismiss this limitation, stating "oftentimes the distributed training assumes some standard fixed budget of workers," this design choice severely restricts the optimizer's applicability in practical scenarios. Particularly, LAgg-A cannot handle partial participation - a fundamental aspect of cross-device federated learning where device availability varies over time [11,12]. This limitation is especially problematic given that partial participation is the default training paradigm in modern federated learning systems [13].

References:

[1] Nesterov, Y. "Introductory Lectures on Convex Optimization: A Basic Course." *Springer*, 2004.

[2] Bottou, L., et al. "Optimization Methods for Large-Scale Machine Learning." *SIAM Review*, 2018.

[3] Stich, S.U. "Local SGD Converges Fast and Communicates Little." *ICLR*, 2019.

[4] Koloskova, A., et al. "A Unified Theory of Decentralized SGD with Changing Topology and Local Updates." *ICML*, 2020.

[5] Woodworth, B., et al. "Local SGD With a Communication Compression Guide." *NeurIPS*, 2021.

[6] McMahan, H.B., et al. "Communication-Efficient Learning of Deep Networks from Decentralized Data." *AISTATS*, 2017.

[7] Li, T., et al. "Federated Optimization in Heterogeneous Networks." *MLSys*, 2020.

[8] Karimireddy, S.P., et al. "SCAFFOLD: Stochastic Controlled Averaging for Federated Learning." *ICML*, 2020.

[9] Li, X., et al. "On the Convergence of FedAvg on Non-IID Data." *ICLR*, 2020.

[10] Hsu, T.M.H., et al. "Measuring the Effects of Non-Identical Data Distribution for Federated Visual Classification." *NeurIPS*, 2019.

[11] Kairouz, P., et al. "Advances and Open Problems in Federated Learning." *Foundations and Trends in Machine Learning*, 2021.

[12] Yang, Q., et al. "Federated Learning: Challenges, Methods, and Future Directions." *IEEE Signal Processing Magazine*, 2019.

[13] Bonawitz, K., et al. "Towards Federated Learning at Scale: System Design." *MLSys*, 2019.
4. Writing quality and mathematical rigor have significant shortcomings:
1) The paper fails to clearly justify its criticism of "hand-designed algorithmic enhancements". For example, in the context of:
```
Work in this field has largely focused on addressing the heterogeneity of data across workers or clients (Karimireddy et al., 2020; Mishchenko et al., 2022). These advancements are generally achieved by hand-designed algorithmic enhancements, whereas our approach relies on more
flexible and potentially more powerful learnable mechanisms that may generalize these and more complex algorithms.
```
The authors don't provide evidence that their approach improves upon established methods like SCAFFOLD or ProxSkip either theoretically or empirically.

2) The paper uses unconventional terminology without proper definition. Terms like "gradient feature" and "handcrafted features" are not standard in the field and remain undefined throughout the text.

3) Mathematical notation lacks clarity, particularly in Algorithm 1. For instance, A_t is introduced in line 8 but is not defined in either Section 3.2 or Appendix A, nor are examples provided for classic algorithms like ADAM or AdaGrad.

4) The definition of $F_\phi$ in Section 4 is ambiguous:
```
Our method builds upon local SGD. After $H$ local steps, we employ a per-parameter learned optimizer $F_\phi$ based on (Metz et al., 2022a) to compute the updated centralized weights (algorithm 1). By computing the centralized update using an expressive neural net $F_\phi$, our method can be seen as a generalization of existing update methods such as taking the average iterate (Stich, 2019) or computing server-side momentum updates (Wang et al., 2019).
```
The paper alternates between describing $F_\phi$ as a "learned optimizer" and an "expressive neural net" without providing a precise mathematical definition.

5) The paper contains imprecise mathematical language, such as referring to "optimizing a model with parameters w" rather than optimizing an objective function.

6) The meta-learning objective function:

$$
\min \phi \mathbb{E}_{\left(\mathcal{D}, w_0\right) \sim \mathcal{T}} \mathbb{E}_{(X, Y) \sim \mathcal{D}}\left(\frac{1}{T K} \sum_{t=0}^{T-1} \sum_{k=0}^{K-1} \mathcal{L}\left(X, Y ; F_\phi(\cdot)\right)\right)
$$

lacks clarity regarding how the loss depends on indices $t$ and $k$, with no rigorous formulation provided even in the appendix.

7) References lack proper formatting, with inconsistent capitalization and missing journal information. Examples include:
```
Qsgd: Communication-efficient sgd via gradient quantization and encoding.
Local sgd converges fast and communicates little
Qsparse-local-sgd: Distributed sgd with quantization, sparsification, and local computations
```

5. Experimental evaluation has several limitations:

1) Scale of experiments is insufficient for modern deep learning:
  While Section 5.5.5 includes experiments with ResNet50 (25M parameters), ViT (5M parameters), and a decoder-only transformer (19M parameters), these architectures are relatively small compared to state-of-the-art models with hundreds of millions or billions of parameters. The paper's primary focus on 2-layer/3-layer MLPs and small CNNs (as evidenced in Table 1) raises concerns about the scalability and practical applicability of the proposed methods to contemporary deep learning models used in research and industry.

2) Absence of standard accuracy metrics:
  The paper deviates from conventional practice in image classification literature by only reporting training loss values. Standard metrics such as Top1 or Top5 accuracy on both training and test sets, which are crucial for assessing practical utility, are notably absent from the evaluation.

3) Limited comparison with relevant baselines:
  While the paper demonstrates improvements over Local SGD in communication-efficient settings, it omits comparison with numerous recent methods that have already shown significant advances over Local SGD/FedAvg [14,15,16,17,18,19,20]. This omission makes it difficult to assess the true practical value of the proposed methods relative to the current state-of-the-art.

4) Visualization and presentation issues:
  Several figures suffer from presentation issues that hinder proper evaluation. Figure 3 lacks logarithmic scaling, making it difficult to assess performance at small loss values. In Figure 2, the similar order of magnitude of losses does not make the experimental contribution seem promising.. These visualization choices limit the paper's ability to effectively communicate the advantages of the proposed algorithms.

References

[14] Karimireddy, Sai Praneeth, et al. "Breaking the Centralized Barrier for Cross-Device Federated Learning." *Advances in Neural Information Processing Systems*, vol. 34, 2021, pp. 28663–28676.

[15] Li, Tian, et al. "Federated Optimization in Heterogeneous Networks." *Proceedings of Machine Learning and Systems*, vol. 2, 2020, pp. 429–450.

[16] Reddi, Sashank, et al. "Adaptive Federated Optimization." *arXiv preprint*, arXiv:2003.00295, 2020.

[17] Mishchenko, Konstantin, et al. "Proxskip: Yes! Local Gradient Steps Provably Lead to Communication Acceleration! Finally!" *International Conference on Machine Learning*, 2022, pp. 15750–15769.

[18] Yuan, Xiao-Tong, and Ping Li. "Sharper Analysis for Minibatch Stochastic Proximal Point Methods: Stability, Smoothness, and Deviation." *Journal of Machine Learning Research*, vol. 24, no. 270, 2023, pp. 1–52.

[19] Yuan, Xiaotong, and Ping Li. "On Convergence of FedProx: Local Dissimilarity Invariant Bounds, Non-Smoothness and Beyond." *Advances in Neural Information Processing Systems*, vol. 35, 2022, pp. 10752–10765.

[20] Karimireddy, Sai Praneeth, et al. "Scaffold: Stochastic Controlled Averaging for Federated Learning." *International Conference on Machine Learning*, 2020, pp. 5132–5143.

---

> ### Author Response · Authors · 2024-12-22
> **General comments and replies to requested changes**
>
> We would like to thank reviewer y11Q for taking the time to review our paper. Your feedback is much appreciated. We are pleased to hear the reviewer believes our use of meta-training in a Local SGD setting is novel, that we have demonstrated our optimizers generalize to larger unseen tasks, and that our empirical evaluation is comprehensive
>
>
> **Convergence guarantees** We note that many effective optimization algorithms developed in the context of deep learning such as Adam and local SGD were initially proposed as heuristic observations without theoretical analysis. L2O methods can pose a particular challenge for guarantees. However, [3.1] provides an avenue for assuring convergence with L2O methods by defaulting to a base optimization algorithm that has convergence guarantees when the learned optimizer does not decrease the loss. This strategy can directly be applied in our setting. We leave this investigation to future work.
>
> **Results for federated learning** Please note that heterogeneous data scenarios are not the problem we try to solve. $We are not federated$. The local SGD setting has applications to training models within datacenters that are too poorly interconnected to yield efficient training when communicating every step. This is the setting we address. Nevertheless, we agree that our method can be trivially applied to federated settings. In section E.6 of the appendix, we have added results applying our glocal learned optimizers to the federated learning settings. Specifically, we meta-train LOpt-A in such settings and show it outperforms many well-tuned baselines including scaffold with respect to test accuracy.
>
> **Regarding criticism of hand-designed optimizers** We disagree that this criticism is insufficiently justified. It is well-known in the machine learning literature that learnable methods outperform hand-designed heuristics (e.g. SIFT descripter v.s. The learned features of a Conv. Net).
>
> **LAgg-A's limited applicability** We acknowledge LAgg-A’s applicability to settings with different numbers of workers K is limited due to the fixed MLP input size. However, we still show strong results for LOpt-A.
>
> **Changes to notation and math** In the main manuscript, we have added a detailed description of Ada features, improved the presentation of algorithm 1, and generally provided more details about our global learned optimizer setup. In the appendix, we have also added an extended description of the meta-training algorithm we use to train LAgg-A and LOpt-A.
>
> **Ablation analysis of different components** We provide such an ablation study for input features in Figure 11 of the appendix.
>
> **Reporting accuracy and training large models** In the main manuscript, we have added an experimental section showing that LOpt-A can be used to train a large-scale task: ResNet-152 on ImageNet, achieving 40% Top-1 Accuracy in 2000 communication steps (~14 epochs), outperforming a tuned SlowMo baseline.
>
> ---
> **Local References**
>
> [3.1][A Simple Guard for Learned Optimizers; ICLR2022]

---

> > ### Comment · Reviewer_y11Q · 2025-01-15
> >
> > Dear authors,
> >
> > Thanks for updating the paper with a new revision and commenting my concerns.
> >
> > While I acknowledge the authors' responses regarding the highlighted weaknesses concerning convergence guarantees and federated learning results, I find several aspects of their rebuttal that require further attention.
> >
> > First, I cannot fully accept their claim about learnable methods consistently outperforming hand-designed heuristics in the machine learning literature (drawing parallels with SIFT descriptors versus learned CNN features). This assertion lacks proper substantiation through academic references or empirical evidence.
> >
> > Furthermore, regarding the issue highlighted by both myself and the Area Editor concerning the primary reporting of training loss, I find the authors' justification citing "standard practice in optimization" to be insufficient. While reporting training loss is indeed common practice, as evidenced in paper [4.1], it is equally standard for optimization papers to provide theoretical guarantees - which this work currently lacks. The absence of such theoretical foundations makes robust empirical validation even more crucial.
> >
> > Given these considerations, I believe stronger practical verification is necessary. Specifically, the paper should:
> > 1. Demonstrate convergence behavior on test datasets
> > 2. Include comprehensive tables with achieved accuracies across different tasks and settings
> >
> > These additions would significantly strengthen the empirical validation of the proposed method, particularly in light of the missing theoretical guarantees.

---

> > > ### Author Response · Authors · 2025-01-17
> > >
> > > **Regarding learnable methods consistently outperforming hand-designed heuristics** Before Deep Learning became popular, handcrafted features were widely used across Computer Vision (CV), Automatic Speech Recognition, Natural Language Processing, and many more domains. While these heuristics are still used today, they are no longer used as part of state-of-the-art (SOTA) solutions. For instance, in 2012 the first deep learning based method, AlexNet [3.3], was applied to win ILSVRC2012 [3.2]. The runner-up submission used SIFT and other hand-designed descriptors for image classification but was outperformed by over 10% top-5 accuracy by AlexNet. Since 2012, all competitive submissions to subsequent ILSVRCs have exclusively used learned features [3.4,3.5,3.6,3.7,3.8]. Similarly, in Automatic Speech Recognition (ASR), MFCC features [3.11] were widely used in SOTA ASR systems [3.12] before Deep Learning became popular. More recent SOTA ASR models use learned features [3.9,3.10]. Natural Language Processing (NLP) is no different. While researchers have traditionally relied on n-grams, word length, character frequency, part-of-speech tagging, dependency relations, Named Entity Recognition (NER), etc to enhance language model performance [3.15], the most recent state-of-the-art models use an end-to-end learned model with learned word embeddings as input [3.13,3.14]. As demonstrated by the transition to end-to-end learned models in CV, ASR, and NLP, there has been a consistent trend within the machine learning literature of moving towards end-to-end learnable solutions instead of relying on hand-designed heuristics. Although there may be some exceptions where learnable methods do not yet outperform hand-designed heuristics, as long as there is enough data to learn from or data can easily be collected, we expect learnable methods will eventually outperform heuristics in these exception cases.
> > >
> > > **Reporting training loss in learned optimization** Please note that "standard practice in optimization" does not appear anywhere in our rebuttal. We do write "standard practice in learned optimization" [4.2,4.3,4.4], which we believe the reviewer is referring to. As such, we disagree that the results in [4.1] constitute common practice for our work. Despite this, please note that we target validation loss in the meta-learning problem in Figure 4 and show strong performance on the validation set. Furthermore, in the rebuttal phase, we demonstrated that LOpt-A can be used to train a large-scale task: ResNet-152 on ImageNet, achieving 40% Top-1 Accuracy in 2000 communication steps (~14 epochs).
> > >
> > > **Theoretical guarantees** Theoretical guarantees are challenging to obtain in the context of learned optimization, but this problem goes well beyond our work which investigates learned optimization in the distributed setting. However, [3.1] provides an avenue for assuring convergence with L2O methods, as we mentioned in our reply above.
> > >
> > > ---
> > >
> > > **Local References**
> > >
> > > [3.2] [https://image-net.org/challenges/LSVRC/2012/results.html]
> > >
> > > [3.3] [ImageNet Classification with Deep Convolutional Neural Networks, Krizhevsky et al., 2012, NIPS.]
> > >
> > > [3.4] [https://image-net.org/challenges/LSVRC/2013/results.php]
> > >
> > > [3.5] [https://image-net.org/challenges/LSVRC/2014/results.php]
> > >
> > > [3.6] [https://image-net.org/challenges/LSVRC/2015/results.php]
> > >
> > > [3.7] [https://image-net.org/challenges/LSVRC/2016/results.php]
> > >
> > > [3.8] [https://image-net.org/challenges/LSVRC/2017/results.php]
> > >
> > > [3.9] [wav2vec 2.0: A Framework for Self-Supervised Learning of Speech Representations, Baevski et al., 2020, NeurIPS. ]
> > >
> > > [3.10] [Robust Speech Recognition via Large-Scale Weak Supervision, Radford et al., 2022]
> > >
> > > [3.11] [Comparison of different implementations of MFCC, Zheng et al., 2001, JCST]
> > >
> > > [3.12] [The Kaldi Speech Recognition Toolkit, Povey et al., 2011, ASRU.]
> > >
> > > [3.13] [BERT: Pre-training of Deep Bidirectional Transformers for Language Understanding, Devlin et al., 2019, NAACL-HLT.]
> > >
> > > [3.14] [Language Models are Few-Shot Learners, Brown et al., 2020]
> > >
> > > [3.15] [Advances in natural language processing, Hirshberg et al., 2015, Science]

---

### Review · Reviewer_QMtu · 2024-10-23

**Summary Of Contributions:**

The paper investigates the potential of learned optimizers, which are meta-learned to perform global updates based on local SGD iterations. These optimizers aim to close the performance gap between local SGD and state-of-the-art adaptive optimizers while maintaining communication efficiency. This work proposes two architectures for learned optimizers: a worker-aware optimizer (LAgg-A) and a worker-invariant optimizer (LOpt-A). These optimizers are meta-learned to aggregate local updates into effective global updates. It is demonstrated that the learned optimizers can generalize to unseen datasets, architectures, and modalities, such as language modeling, when meta-learned on a single or few combinations of architectures and datasets.

**Audience:**

Yes

**Claims And Evidence:**

Yes

**Requested Changes:**

Major revision

**Strengths And Weaknesses:**

Strengths:

1.The topic studied is meaningful for improving the model training efficiency.

2.Extensive experiments are conducted to verify the effectiveness of the proposed methods.


Weaknesses:

1.Please highlight the contributions and the novelty of the paper. It seems that this work is a direct and simple combination of local SGD and meta learning. There is no special design for the two optimizers, LAgg-A and LOpt-A.

2.Could you please compare the total training time of different optimizers in the experiments.

3.Besides the losses during training, the model performance (e.g., accuracy) is expected to be compared. In some cases, the optimizer that achieves fast convergence rate can result in degraded performance.

---

> ### Author Response · Authors · 2024-12-22
> **General comments and replies to requested changes**
>
> We would like to thank reviewer QMtu for taking the time to review our paper. Your feedback is much appreciated. We are pleased that the reviewer believes our work is important for improving model training efficiency and that we have conducted extensive experiments.
>
>
>
>
> **Timings for our leaned optimizers** We have addressed this concern in the general reply.
>
> **Including Accuracy** In the main manuscript, we added an experimental section showing that LOpt-A can be used to train a large-scale task: ResNet-152 on ImageNet, achieving 40% Top-1 Accuracy in 2000 communication steps (~14 epochs).
>
>
> **Regarding Novely** To the best of our knowledge, the distributed low-communication setting has not previously been studied in the learned optimization literature, despite it being an interesting setting. Not only does learned optimization have the potential to greatly improve the efficiency of Local SGD, but the Local SGD framework itself is appealing for learned optimization.
>
> Learned optimization is an appealing solution for Local SGD as it can allow learning more effective strategies to deal with the client drift problem, thus making the method more communication efficient. In traditional LocalSGD, it is difficult to perform well when H, the number of communication steps, grows to be large [2.4,2.5]. However, our method shows strong performance and much faster convergence than baselines at higher H (Figure 3, Table 2).
>
> The Local SGD framework is also appealing for learned optimization for two reasons. First, it allows to meta-train the learned optimizer for fewer total steps while still improving optimization performance. Improving the generalization of the learned optimizer (LO) to longer unrolls (more total steps) of the inner problem is a crucial open problem in learned optimization [2.1,2.2], however, in Local SGD, the total number of steps for the inner problem naturally shrinks by a factor H. Therefore, training LOs in the Local SGD setting naturally alleviates the unroll problem. Second, the overhead of the learned optimizer shrinks to 0 as the number of local steps H grows (see table 1). In communicate-every-step settings learned optimizers can have a meaningful overhead compared to hand-designed optimizers [2.3], however, this need not be the case for Local SGD. Therefore, training LOs in the Local SGD setting can help reduce a drawback of LOs.
>
> In summary, learned optimization helps alleviate the client drift problem of Local SGD and meta-training LOs in the local SGD setting allows to amortize the LO cost over H local steps and alleviates the problems of LOs for generalizing to longer unrolls. As such, the complementary nature of global learned optimizers goes beyond a simple combination of Local SGD and LOs since the new framework has benefits that were not present in either setting before.
>
>
> ---
> **Local References**
>
> [2.1] [VeLO: Training Versatile Learned Optimizers by Scaling Up, Metz et al.]
>
> [2.2] [Harrison et al. A Closer Look at Learned Optimization: Stability, Robustness, and Inductive Biases, Neurips 2022]
>
> [2.3]  [Practical Tradeoffs Between Memory, Compute, and Performance in Learned Optimizers, Metz et al., CoLLAs 2022]
>
> [2.4] [Local SGD Converges Fast and Communicates Little;  ICLR 2019]
>
> [2.5] [​​SlowMo: Improving Communication-Efficient Distributed SGD with Slow Momentum; ICLR 2020]

---

### Review · Reviewer_3zHo · 2024-11-19

**Summary Of Contributions:**

The paper presents a new approach for distributed learning that is a variant of local SGD of Stich (2019). In local SGD, workers take several local gradient updates, communicate the cumulative parameter diffs (deltas) to the server, who then averages the deltas to update the weights. Instead of vanilla averaging, the paper instead proposes using a learnable function that maps deltas (and a list of auxiliary "Ada" scalar features that are derived from the deltas) to updates. This learnable function can be a shallow MLP, and can be meta-learned by optimizing over a distribution of tasks and datasets.

The proposed method is validated via a range of experimental results and is shown to improve upon local SGD. In particular, the optimizer shows generalization abilities to larger architectures, bigger datasets, and even new modalities (such as language model training).

**Audience:**

Yes

**Claims And Evidence:**

Yes

**Requested Changes:**

* Please move the description of Ada features to the main paper, preferably by fleshing out Section 3.2 in greater detail.
* Please address (and attempt to fix) the questions listed in the weaknesses above.

**Strengths And Weaknesses:**

I enjoyed this paper. The authors do a good job of motivating the method, describing its particulars, and benchmarking performance. The combination of meta-learning and local SGD is intuitive. I especially liked the clarity of the experimental sections.

However, there are some issues which I would suggest having addressed before publication:
* For completeness it will be helpful to include the full pseudocode for the meta-training algorithm (along with details such as hyperparameters) in the main paper. I can probably reconstruct some of it by reading Appendix A but am not certain.
* For completeness it will be helpful to include the full description of the Ada features in the main paper.
* $u_t$ --- the accumulator variable? -- is used in Algorithm 1 but not defined in the main paper (as far as I can tell).

Some other minor comments:
* I didn't quite understand the phrase "shared steps are not colored" in the algorithm description. Do you mean "shared steps are colored"?
* It may be helpful to reflect more in detail what exactly the learned optimizer looks like. (Interpreting MLPs can be hard, but for example -- is it very different from just averaging the gradients (like local SGD does)? Are any particular features it is picking up on? Do inductive biases matter? etc.

---

> ### Author Response · Authors · 2024-12-22
> **General comments and replies to requested changes**
>
> We would like to thank reviewer 3zHo for taking the time to review our paper. Your feedback is much appreciated. We are particularly pleased to hear that you enjoyed the paper, believe that the use of meta-learning for Local SGD is intuitive and that our results are clear.
>
>
> **Ada features in main paper** We have added a thorough description of the Ada features in the main text and have additionally fleshed out sections 3 and 4 in greater detail.
>
> **Adding the Full Meta-training Algorithm** We have added a full description of our meta-training algorithm in the appendix. We would be happy to add it to the main paper if the reviewer believes it is necessary given the additional information now available in sections 3 and 4.
>
> **Algorithm 1 caption** We have modified the caption of algorithm 1 to disambiguate the shared steps.

---

### Author Response · Authors · 2024-12-22
**General reply to all reviewers**

We would like to thank all the reviewers for taking the time to review our paper. We are pleased to hear that 3zHo enjoyed reading our paper and that 3zHo and y11Q  believe the use of meta-training in a Local SGD setting is intuitive and novel. We appreciate that QMtu believes our work is important for improving model training efficiency. Finally, we are glad that y11Q and QMtu believe the experiments we conducted are comprehensive and extensive, respectively.

**Timings for LAgg-A and LOpt-A v.s. Local SGD** In the main manuscript, we have added new experiments in a distributed setting comparing timings for LAgg-A and LOpt-A to Local SGD, showing that the overhead of our optimizers over Local SGD tends towards 0 as H increases.

---

### Comment · Action_Editor_2gXp · 2024-12-23
**More questions from AE**

Dear authors,

Thanks for updating the paper with a new revision and answering some reviewers' questions and concerns. I also did a pass over the paper and have several questions (some of which were also raised by the reviewer y11Q). Could you clarify these for me?
- (major issue) First, why do you report mainly train losses only in the main paper? I think it is insufficient to justify anything as there could be strong overfitting and thus I suggest you to monitor either both train and validation loss, or report only validation loss. And then yep, report per task its metric, e.g. for classification it is accuracy.
- I found it confusing that you reference local SGD as the work of 2019 paper, while it is exactly fedAvr introduced in 2016. Maybe it is good to discuss a bit more differences (at least on the application side) from both papers first and make clear maybe the initial reference on local SGD. (I could be wrong here in terms of who first introduced the averaging exactly, but at least make discussion to avoid confusion)
- I think it is not very clear from the text right now on the setting you are considering exactly. It would be nice to clearly state that it is homogeneous devices, data uniformly splitted between devices, devices has specific limitations on disk and memory (e.g. it is not phones but data centers with machines, thus huge amounts of data can be stored on one device). This is critically important as e.g. if you check DiLOCO paper https://arxiv.org/abs/2311.08105 and federated learning results -- settings are very different which led to different empirical observations on the optimization. Right now I see it is not discussed in depth.
- (major issue) If you are targeting the regime of data parallel training with data centers or at least GPU / TPU machines (not general decentralized training in federated learning with limited memory and compute bound devices, with heterogeneous data and issues with connection / async training), then why did you consider so small models of sizes < 30M? This is really setting the federated learning most works target as you have limitations on the memory / compute per device. In case of datacenters I cannot imagine practical scenario right now of training such small data to optimize really communication (as we train not on big data here obviously + the model size to communicate is really super small and mainly we are data bound right now for this kind of models with the current hardware we have in community - data loader is slow compared to grad computation and synchronization for small models).
  - if you wanna make a statement for more realistic federated learning - appendix results should be moved to the main paper, and moreover in depth comparison should be done: e.g. why did you use validation loss and not accuracy to select best hyperparams? results on CIFAR-10 look poor -- accuracy is way lower than e.g. reported by Reddi in FedAdam.
  - if it is not the goal, and you still target improving general data parallel training with data centers, then you need to compare with DiLOCO e.g., but more important for more realistic model sizes and data sizes (e.g. ImageNet is ok, but not CIFAR or MNIST).
- (major) If I assume that validation loss behaves similar to train loss, then all your plots are showing that local SGD converges better at the same communication step than SGD. This is very surprising as we know that in general SGD should converge better than local SGD and moreover if converges often faster than local SGD. I don't understand this result. Probably the issue is usage of K × H × B_loc as the batch size for the SGD -- in this case it is actually doing the same number of synchronization steps as local SGD. I think this is a wrong baseline in general, you should consider K × B_loc batch size for SGD and perform T × H steps of optimization and compare both methods by plotting loss / accuracy with x-axis being the number of samples model trained on.
- VIT is a transformer and often it is the issue to train transformers with SGD to have good and stable performance. How were you able to train it at all? Is there any need for adaptive optimizer and also learning rate warmup?

AE.

---

> ### Comment · Action_Editor_2gXp · 2025-01-02
> **Any comment?**
>
> Dear Authors,
>
> Any comment for the above questions I still have?
>
> Thanks,
>
> AE.

---

> ### Author Response · Authors · 2025-01-03
> **Reply to AE**
>
> **Regarding reporting training loss** Reporting validation loss is orthogonal to the focus of our paper. Optimization methods are typically designed to optimize the training loss, whereas generalization to a test set is studied separately (e.g. Adam was initially proposed and only later on was it found to overfit in some settings unless its weight decay is controlled properly [4.12]). With this in mind, we follow standard practice from prior published works in learned optimization [4.2,4.3,4.4], which focus on studying training loss for simplicity in evaluating optimizers. This choice is discussed and clarified in the introductory paragraph of section 5.
>
> In L2O, generalization is commonly addressed by adding weight decay [4.3], targeting validation loss [4.4] (see our Fig 4) or adding extra regularizers [4.5]. For example, we target validation loss in Figure 4 and show strong performance on the validation set. Furthermore, in the rebuttal phase, making use of weight decay, we demonstrated that, despite only targeting training loss, LOpt-A can be used to train a large-scale task: ResNet-152 on ImageNet, achieving 40% Top-1 Accuracy in 2000 communication steps (~14 epochs). As such behavior is expected in the optimization literature, including L2O [4.3,4.5], it is appropriate to report only the training loss behavior, since as we mentioned above improving on the test set is often orthogonal and can be a confounding factor making the analysis more complicated.
>
>
> **The setting we consider and Fedavg vs local SGD** The setting we consider is exactly that of homogeneous devices and with the data not being federated, meaning that each worker can draw from the same pool of data iid [4.1,4.10]. We have added text stating this explicitly in the abstract, introduction, before the contributions, in the experiments section, and in the conclusion. We acknowledge that [4.6] (cited in our paper) introduced federated averaging and that the algorithm directly applies to the homogeneous setting we consider. However, given that our focus is not federated learning (we have only added additional experiments in the rebuttal phase due to y11Q’s request), we mainly refer to [4.1 and 4.10] as the previous work throughout the paper. We have clarified this in paragraph 2 of the introduction.
>
>
> **Why do we experiment with smaller models** Meta-training learned optimizers is expensive and generalizing from small meta-training tasks to large tasks is still an open problem for L2O. Therefore, to show larger-scale experiments, we would need to meta-train on large tasks which is currently out of reach given our academic resources and beyond the scope of this initial investigation. With this in mind, in our paper, we focus on demonstrating that our global learned optimizers possess analogous meta-generalization abilities to communicate-every-step learned optimizers in the literature [4.2,4.3,4.4].  That is, we show that the powerful framework of learned optimization can be applied to communication-efficient distributed learning with success. We leave the creation and benchmarking of a more general global learned optimizer (e.g. as was done for VeLO [4.9] in the communicate-every-step setting) to future work.
>
>
>
> **Comparison with DiLOCO** DiLOCO is a concurrent and unpublished work focusing on LLMs. The main technical difference is using AdamW as the inner optimizer, which is complementary to our method as it just involves changing the inner optimizer and can trivially be combined with our method. Therefore, we leave the incorporation of DiLOCO within our framework to future work.
>
>
> **SGD convergence** Based on prior works in our setting [4.10, 4.11], the Adam and SGD baselines in our experiments assume a data parallel setting where K workers process batch sizes that are the same as the H*B_local (e.g., baseline A3 from [4.10; Figures 1]). We recall that local sgd was introduced as an alternative to larger batch sizes in [4.10]. Therefore, these baselines match the training time and total communication volume of local SGD. We acknowledge that reducing the batch size to K * B_loc and taking H * T total steps could lead to better performance in some cases, as reported in [4.10; Figure 1 baseline A2]. However, the time-to-accuracy will also increase in this case [4.10; Table 1]  see k=8, h=1 vs h=8. Given that our focus is on communication-efficient distributed settings and that prior work makes such tradeoffs clear, we chose not to include baselines that require more communication or wall-clock time than local SGD.

---

> ### Author Response · Authors · 2025-01-03
> **Reply to AE continued**
>
> **FedAdam Results** During the rebuttal phase, we have provided entirely new federated learning results that are of interest to the community, but tangential to our main focus on homogeneous devices without data federation. We target validation loss here to tune FedAdam since we use validation loss to meta-train our learned optimizers (hyperparameter tuning is analogous to meta-training). Please note that our federated learning study is meant to be standalone; all optimizers within it are comparable to each other. We did not take explicit steps (e.g. incorporating the same data augmentations, etc.) to match the performance reported by [4.7].
>
> **Training ViT with SGD** SGD can train ViT [4.8; Figure 10 (a) ]. While Adam certainly outperforms SGD for training transformers [4.8], it would be incorrect to say SGD cannot train transformers. Please note that our experiments show SGD in the data-parallel setting, that is with a batch size = k*h*B_loc. This large batch size may be helping to stabilize training with SGD.
>
>
>
> **Local References**
> ---
>
> [4.1] [Local SGD Converges Fast and Communicates Little;  ICLR 2019]
>
> [4.2] [Practical Tradeoffs Between Memory, Compute, and Performance in Learned Optimizers, Metz et al., CoLLAs 2022]
>
> [4.3] [Harrison et al. A Closer Look at Learned Optimization: Stability, Robustness, and Inductive Biases, Neurips 2022]
>
> [4.4 [Understanding and correcting pathologies in the training of learned optimizers, ICML 2019]
>
> [4.5]  [Junjie Yang et al., Learning to Generalize Provably in Learning to Optimize, AISTATS 2023]
>
> [4.6] [McMahan et al. Communication-Efficient Learning of Deep Networks from Decentralized Data, AISTATS 2017]
>
> [4.7] [Adaptive Federated Optimization, ICLR 2021]
>
> [4.8] [Why Transformers Need Adam: A Hessian Perspective, NeurIPS 2024]
>
> [4.9] [VeLO: Training Versatile Learned Optimizers by Scaling Up, Metz et al.]
>
> [4.10] [Don't Use Large Mini-Batches, Use Local SGD; ICLR 2020]
>
> [4.11] [​​SlowMo: Improving Communication-Efficient Distributed SGD with Slow Momentum; ICLR 2020]
>
> [4.12] [Decoupled Weight Decay Regularization; ICLR 2019]

---

### Author Response · Authors · 2025-03-03
**We would like to thank the editor and reviewers for their constructive feedback on our paper throughout this process.**

We have now uploaded our final changes to the manuscript:
- We include test accuracies (or test loss where applicable) for all results in the main figure: ResNet50 training on ImageNet64, ViT training on ImageNet64, MLP training on ImageNet32, and a transformer language model trained on LM1B.
- We add another SGD baseline to quantify the performance one can expect from taking KxHx1000 steps with B_loc batch size, instead of 1000 steps of local-SGD. We include this baseline in the main figure (Fig.2), the H ablation figure  (Fig.3), and the K ablation figure (Fig.10). We also added details of our hyperparameter search in the appendix (Table 6).
- We have added a discussion of DiLoCo in the related work section 2.1 and in the results section 5.3.
- We improved the writing and presentation in the appendix.

---

> ### Comment · Action_Editor_2gXp · 2025-03-10
> **Questions about SGD baseline and test accuracy**
>
> Dear Authors,
>
> Thanks for updating the manuscript and including all things I requested. Have several clarifications questions based on the revision:
> - Fig 2, 3, 10 another SGD baseline is orange and orange with dash-dot is the final performance of that SGD? I think would be good to add this into the caption or text to be clear.
>   - Now you could see that some optimizations could be better :) this is what I wanted to see, and also local SGD is worse which is consistent with prior works.
> - why is test accuracy in Figure 2 so low? Maybe typo? it is less than 10% so I would say all models then are really useless from the generalization perspective.
>
> The rest looks good to me! Thanks,
>
> AE.

---

> ### Author Response · Authors · 2025-03-17
> **Reply to AE**
>
> Dear AE,
>
> Thank you for raising these concerns. We have replied below and updated the manuscript accordingly.
>
> **Captions.** We have now updated the manuscript regarding the figure captions.
>
> **Regarding Low accuracies in Figure 2.** MLPs are expected to converge poorly on ImageNet. For ViT and ResNet50, the optimizers see B_loc * H * K * 1000 images, corresponding to approximately 3.2 epochs of ImageNet training similar to [4.2]. Therefore, the results in Figure 2 for ImageNet training are far from convergence. Moreover, we use resized images (see Figure 2’s caption), leading to lower performance than full-sized images.  We meta-train our learned optimizers for 1000 steps, thus evaluating them outside of this task horizon would lead to an unfair comparison [4.2]. Improving generalization to longer unrolls is a challenging open research question in learned optimization [4.3]. Since we do not focus on generalization to longer unrolls, we evaluate our optimizers exclusively within the meta-training horizon. Please note that, as we mentioned earlier in our rebuttal, “to show larger-scale experiments, we would need to meta-train on large tasks which is currently out of reach given our academic resources and beyond the scope of this initial investigation.” Finally, during the rebuttal phase, we added Figure 6, showing reasonable accuracy with longer training (~14 epochs) and a better optimizee architecture. To further clarify the convergence of models in Figure 2, we have added a paragraph about it in section 5.1.3 and included an explanation in Figure 2’s caption.
>
> **Figure 2 (g,h).** When reviewing the results from Figure 2, we identified a minor discrepancy (the batch size was incorrectly set) with our Adam baseline impacting only the LM1B results in subfigures (g,h). We have now updated the results to reflect the correct batch size.
>
> Best regards,
>
> Authors

---

> > ### Comment · Action_Editor_2gXp · 2025-03-18
> > **Reply**
> >
> > Dear Authors,
> >
> > Thanks for the update and clarifications! All good from my side!
> >
> > AE.

---

### Decision · Action_Editor_2gXp · 2025-02-02

**Recommendation:** Accept with minor revision

**Comment:**

Overall, making decision for the paper was hard due to several reasons:
- absence of strong theoretical guarantees while performing empirical analysis similar to prior works and comparing only training loss.
- comparison with the strong prior baseline on SlowMo, while usage of specific settings for SGD baseline.
- small-scale experiments due to limited resources and expensive / unknown better optimization of meta-learning while large scale experiments are needed for empirical strong support.

Conceptually, I believe that we need to improve validation of results from the prior works to improve applicability of empirical / theoretical finding of optimization methods by comparing validation loss / accuracy, using large models and datasets, considering different variants of the SGD baseline training.

At the same time having TMLR guidelines, I believe that
- justification for small scale experiments authors provided are acceptable, so this should not affect the final decision if provided results are correct and useful for the community (e.g. paper definitely can serve as an initial step for the following works to properly select how to perform scaled version of analysis).
- comparison with prior works on the communication efficient optimizers is valid, thus the SGD baseline itself did not diminish authors’ contributions and the main story they tell.
- while strong theoretical guarantees are intractable, well established empirical analysis serves as a strong piece of work, as TMLR welcomes both empirical and theoretical results.

As some reviewers and I (AE) conceptually believe that empirical results miss the validation loss in the light of absent theoretical guarantees, I understand the authors' point that prior works and optimization theory itself mainly consider the train loss leaving generalization aside. The paper itself brings interesting results and insights and for the cases where the validation performance was provided it looks even stronger from a practical point. Having all contributions in the paper and strong effort on ablations even those which are orthogonal to the work central topic (e.g. federated learning), I believe community will benefit from the work as a starting point to continue unifying local SGD and federated learning research clearly considering different regimes of data / model / capacity of device / number of devices and showing differences in optimization between these regimes.

However, I request authors to include the following changes in the final revision (which I believe minor and can be done):
- Please include test accuracies in the main paper results to boost practicality of the proposed optimizers and drive usage of them in practice where people are interested in generalization. Based on the results from the paper, I believe all your test accuracies should be in line with the training loss.
- DiLOCO paper was published at ICML 2024 workshop, so it is not fair to say that it is concurrent work (as the first version appeared earlier). I agree that comparing with DiLOCO is not needed, as their proposal is orthogonal to meta learning proposed by authors + it was tested only on LM task, not vision. However, I encourage authors to include some general discussion in the related works comparing with DiLOCO and e.g. discussing how many local steps is possible to do in both works.
- Please add another SGD baseline to quantify exactly performance people can expect if they want to switch from regular training to a communication efficient one.
	- While I understand that authors follow the prior works on that baselines, I believe for practical applications we need to have the baseline standard in the community w/o communication efficient optimization.
	- While I got the point of authors that they consider SGD baseline with the same wall-clock time, having complexity of optimization with very large batches, we do not train them in practice (or do some tricks to be able to do it, e.g. progressive batch scaling, etc.) and thus training longer but with smaller batch (ending up with ~total same compute, but longer wall-clock) gives better results. Also for the setting authors consider (matching the training time and total communication volume of local SGD) SGD is trained with H*B_local having H steps of grad accumulation. This means that setting of SGD with B_loc batch but KxH steps will be slow only by communication cost which happens every step instead of once per H steps as in local SGD. There is always the trade off between compute, wall clock and performance as authors mentioned. However, proposing a communication efficient method one could expect to see the same performance as before but now with reduced wall-clock. But absence of that SGD baseline (training KxH steps with B_loc) cannot help to answer the question if one gets the same performance as before while having speedup in wall clock due to reduced communication by H times.

**Audience:**

The paper is targeting communication efficient SGD, which means that any person in the community can benefit as we are moving towards reducing the cost of training. The paper also shows comparison as ablation between federated learning and DDP training if the meta-learning is used to improve communication efficient SGD. The paper has a lot of empirical results at small scale, which can serve as the foundation for future works on meta learning to improve optimization efficiency for model training.

**Claims And Evidence:**

The paper targets improvements of communication-efficient distributed training optimization by suggesting to use meta-learning to optimize the aggregation of local updates. First of all, authors propose a new way to perform aggregation of local updates via meta-learning, LAgg-A and LOpt-A, which is applicable to any local optimizations, including local SGD. Second, authors provide many empirical results considering different datasets and models and comparing meta-learning with prior communication-efficient methods, showing that meta-learning outperforms prior variations of local SGD. Authors also analyse how different parameters of the method and local optimization influence model training, including federated learning setting, compression, ada features, in-distribution/out-of-distribution, number of workers / local steps, etc.

The paper lacks theoretical guarantees, while providing comprehensive empirical results. Several concerns were raised regarding the baselines and reporting final results solely based on training loss, as well as how to do extension to federated learning regime and partial participation. The work was revised to be focused exactly on the regime of homogeneous data and DDP variant of training, leaving the federated learning regime as an ablation and possible future work.